# Using evolution to generate sustainable malaria control with spatial repellents

Penelope Anne Lynch[1]*, Mike Boots[1,2]

[1]Department of Biosciences, University of Exeter, Cornwall Campus, Penryn, United Kingdom; [2]Department of Integrative Biology, University of California, Berkeley, United States

**Abstract** Evolution persistently undermines vector control programs through insecticide resistance. Here we propose a novel strategy which instead exploits evolution to generate and sustain new control tools. Effective spatial repellents are needed to keep vectors out of houses. Our approach generates such new repellents by combining a high-toxicity insecticide with a candidate repellent initially effective against only part of the vector population. By killing mosquitoes that enter treated properties the insecticide selects for vector phenotypes deflected by the repellent, increasing efficacy of the repellent against the target vector population and in turn protecting the insecticide against the spread of insecticide resistance. Using such evolved spatial repellents offers an evolutionarily sustainable, 'double-dip' system of disease control combining mortality and repellence. We formalize this idea using models which explore vector population genetics and disease transmission probabilities and show that using evolved spatial repellents is theoretically achievable, effective and sustainable.

## Introduction

Vector-borne diseases including malaria and dengue remain a major burden to human health despite decades of funding targeting their eradication. One child per minute still dies of malaria globally (*WHO, 2015a*), and dengue infects an estimated four hundred million individuals per year (*Bhatt et al., 2013*; *Murray et al., 2013*). One of the key reasons for this has been the impact of evolution on our control tools, with both theory and experience pointing to the inevitability of resistance to conventional insecticides (*Brogdon and McAllister, 1998*; *Hemingway et al., 2002*; *Nauen, 2007*; *Asidi et al., 2012*; *Hemingway and Ranson, 2000*; *Ndiath et al., 2012*). It is clear that mosquito populations have considerable potential to evolve and that to achieve sustainable control, it would be prudent to develop approaches that take evolution into account. Very fundamentally, there is a need for evolutionarily rational disease intervention (*Boots, 2015*). Here, we highlight a potential intervention strategy in which evolution in the vector may be utilized to improve rather than reduce disease control. Specifically, we argue that combining initially limited repellents with toxic insecticides has the potential to select for effective repellence and to lead to evolutionarily sustainable control using evolved spatial repellents ('ESR's).

It is important to understand that substances can affect vectors in multiple ways (*Kennedy, 1947*; *Miller et al., 2009*) and there is considerable overlap in the terms used to describe their different actions. We adopt the terminology of Greico et al. (*Grieco et al., 2007*) who characterize three modes of action for the insecticides used in vector control programs; toxicity, contact-irritance and spatial repellence. Toxicity is simply the capacity to generate mortality in mosquitoes that contact the insecticide. Contact-irritance is the repellence of mosquitoes when they contact the insecticide, and spatial repellence acts at a distance, deflecting mosquitoes before they contact a treated surface. Specifically, we are considering repellence that acts to protect people indoors by preventing

*For correspondence: P.A. Lynch@Exeter.ac.uk

**Competing interests:** The authors declare that no competing interests exist.

**eLife digest** Many of the mosquito species that transmit malaria have evolved to bite humans indoors at night, and therefore health programs target them using insecticides sprayed on surfaces inside people's homes. This strategy, however, stops working when mosquito populations evolve to resist the insecticide used, either because they are immune to its poisonous effects or because they change their behaviour to avoid it. Consequently, there is now a need to develop alternative strategies to control mosquitoes that are more sustainable in the face of evolution. One possibility is repellents that keep mosquitoes out of homes.

Lynch and Boots have now asked whether evolution could be used to create effective repellents from substances that initially repel only part of the mosquito population by pairing them with lethal insecticides sprayed inside people's homes. Mathematical models showed that, before insecticide resistance becomes widespread, this "evolved repellence" approach could reduce the spread of malaria by a similar amount to using insecticides alone. This was particularly true if the models considered that, as well as surviving to give fewer infectious bites, repelled infectious mosquitoes may be less likely to transmit malaria with each feed, for example if they feed more on livestock rather than humans.

The models of Lynch and Boots also show that that the success of the evolved repellence concept in a given location depends on a number of factors. The proportion of the starting mosquito population that is repelled or resistant can have a large effect. Similarly, success will also depend on how likely normal, repelled and insecticide-resistant mosquitoes are to reproduce successfully. These values can be influenced by the choice of insecticide and repellent and how the chemicals are applied. Lynch and Boots show that swapping insecticides can allow an evolved repellent to be established where it would otherwise not succeed. Also, the spread of resistance to the paired insecticide is slowed or prevented when the mosquito population evolves to be repelled.

Practical laboratory and field- work is now needed to build on this theoretical groundwork and to determine suitable locations and application strategies to exploit this concept as a way to sustainably reduce the spread of malaria in the future.

malaria vectors from entering treated properties, rather than more localized repellent effects that might deflect them away from individuals outdoors or from treated surfaces indoors. The two approaches are distinct, localized repellence from treated bednets, for example, provides protection only for individuals with nets, and only at times of day when they are under the nets.

In the absence of resistance, DDT, the most effective chemical used to date for the control of malaria vectors, has both high toxicity and high spatial repellence (*Kennedy, 1947*; *Grieco et al., 2007*). Work inspired by the historic efficacy of DDT as a tool in combating malaria has highlighted the potential role of spatial repellence as a means of reducing transmission (*Achee et al., 2012*; *Loyola et al., 1990*; *Roberts et al., 2000*; *Roberts, 2010*; *Roberts et al., 2000*; *Curtis, 2002*; *Roberts and Alecrim, 1991*). The malaria vectors targeted by public health campaigns using bednets and indoor residual spraying of insecticides (IRS) typically feed indoors between dusk and dawn. Vector control has therefore focused on exploiting this behaviour to deliver lethal control measures against indoor-feeding mosquitoes (*Yakob et al., 2011*; *WHO, 2015b*), with outdoor biting viewed as unwanted behavioural resistance (*Bradley et al., 2012*; *Russell et al., 2011*; *Reddy et al., 2011*; *Cooke et al., 2015*; *Mouchet et al., 1963*; *Gatton et al., 2013*; *Killeen et al., 2011*). However, it is increasingly being acknowledged that a switch away from indoor biting, by reducing vector fitness and/or by creating a less favourable context for transmission, may offer public health benefits as an end in itself, consistent with the idea that the efficacy of DDT may be partly derived from its repellent rather than solely its toxic effects (*Roberts et al., 2000*; *Roberts, 1993*; *Grieco et al., 2000*). The use of spatial repellents as a tool to maintain vector-free homes is therefore now being actively investigated as a means of reducing transmission of malaria and dengue (*Achee et al., 2012*; *N'Guessan et al., 2006*; *Ogoma et al., 2014*; *University of Notre Dame, 2016*). This can be viewed as an alternative means to achieve the same benefits as the application of physical barriers to prevent vector access to homes (*Roll Back Malaria, 2015*; *Patrick, 1900*;

*Menger et al., 2016*). Syafruddin et al. have conducted a proof-of-concept double-blind placebo-controlled field trial using spatial repellents disbursed from burning coils. This trial recorded an approximately 50% reduction in malaria infections in repellent-protected properties (*Syafruddin et al., 2014*). They emphasize the need for repellents able to achieve the same effects without combustion in order to develop practical public health interventions. Given the benefits of insect repellence in terms of nuisance reduction as well as disease control, and the consequent long-standing interest in identifying safe and effective repellents, the limited choice and efficacy of currently available compounds suitable for this purpose suggests that finding suitable new repellent substances is a particularly challenging task. Although many promising potential repellents are being evaluated for personal protection, for localized outdoor and indoor protection, and for the property-scale protection relevant for ESRs (*Syafruddin et al., 2014*; *Govella et al., 2015*; *Abiy et al., 2015*; *Revay et al., 2013*; *Müller et al., 2009*), sustained, effective action at a distance is still challenging to achieve, and still often requires active dispersal through combustion or powered devices (*Revay et al., 2013*; *Müller et al., 2009*). DDT itself carries a reputational burden from its historic overuse, such that environmental concerns make its continued use problematic and its continued availability uncertain.

We propose that for the vectors currently targeted by IRS campaigns, there is an opportunity to exploit the evolutionary processes generated by using lethal insecticides to create effective new spatial repellents from compounds that initially repel only a fraction of the vector population. Specifically, we propose a combination of a spatial repellent, which deters mosquitoes from entering buildings, with a low-contact-repellence high-toxicity insecticide, which kills those which do enter, leading to a 'double-dip' system of disease control. If failure to enter buildings is seen as a method of transmission reduction in its own right, then the repellent provides transmission reduction by deflecting mosquitoes. For mosquitoes that are not deflected, the mortality imposed by contact with the insecticide will provide transmission reduction in the same manner as current insecticide-only IRS control methods. Provided that, in a given context, the fitness cost of being deflected is less than the fitness cost of being susceptible to the insecticide, there is potential for selection to favour an increase in the proportion of deflected individuals in the treated population. If deflection is viewed as a form of 'behavioural resistance', this system actively exploits 'resistance' evolution, as mortality generated by the insecticide serves to select for phenotypes that are deflected by the repellent. Candidates for use as spatial repellents thus only need initially to repel a small proportion of a mosquito population, since in this instance, for once, evolution will work to enhance the efficacy of a disease-control measure. Here, we explore the feasibility of this approach using evolutionary models that explore a wide range of possible vector characteristics and disease parameters.

## Modeling

If deflecting mosquitoes from accessing humans indoors is in itself an effective means of reducing transmission, as proposed by *Achee et al. (2012)*, then selection for mosquitoes which are repelled from treated houses could serve to generate new public health tools. Critically, selection would depend on the relative fitness of mosquitoes that are deflected away from buildings compared to those entering buildings. This in turn would depend on the proportion of properties treated with insecticides, the susceptibility or resistance (physiological) of mosquitoes contacting indoor insecticides and the fitness costs associated with being deflected away from the sleeping indoor hosts which vector species have evolved to exploit. However, the fitness costs of being deflected from human dwellings are difficult to determine directly. Following a well-established history of mathematical modeling to explore issues relating to the evolution of resistance in malaria vector populations (for example [*Rosenheim and Tabashnik, 1990*; *Le Menach et al., 2007*; *Mandal et al., 2011*; *Georghiou and Taylor, 1977*; *White et al., 2014*]), we have therefore developed an analysis to explore the possible outcome across a range of fitness scenarios, using a feeding-cycle based, two-locus, bi-allelic population genetics model, capturing the mosquito life-history characteristics of one initial mating and pre-adult development period as well as the *Plasmodium* development period in infected mosquitoes before transmission is possible.

As well as the spread of resistance and deflection alleles over time, our model also tracks adult population size and the expected number of infectious bites given by the population during each modeled time unit. Each modeled time unit corresponds to the length and reproductive outcomes of a single feeding cycle, an approach used in previous models of vector population genetics

(*Lynch et al., 2012*; *Read et al., 2009*). For details of the model see Appendix 1. To minimize the sensitivity of the model results to specific parameter values, we frame our key disease control results in terms of the proportionate difference between the model's calculated infectious bite values for a given set of intervention assumptions and those assuming no intervention, minimising the impact of parameter values that are unaffected by the intervention. The impact of deflection on malaria prevalence is determined by the proportion of mosquitoes deflected by a repellent and the probability (compared to non-deflected mosquitoes) that they will then acquire and transmit a *Plasmodium* infection. The effect on *Plasmodium* transmission of deflecting vectors to outdoor biting has not been definitively measured in the field, we therefore consider a wide range of per-feed probabilities of *Plasmodium* transmission to deflected mosquitoes compared to the 4% probability assumed for indoor feeds. Reductions in this parameter are intended to represent the effects of all potential sources of reduced transmission to the vector, including deflection to non-human and therefore non-infectious hosts. In terms of model results, proportionate reductions in this parameter will have the same effect as proportionate changes in the probability that an infectious mosquito which survives to feed will give an infectious bite to a human host. As such, the reductions explored can be interpreted as the product of the proportionate reductions in transmission to and from outdoor-feeding vectors.

Our key model assumptions. (1) Physiological insecticide resistance is controlled by a single locus bi-allelic autosomal gene, with the resistance allele being completely dominant to the susceptibility allele. (2) Deflection by a given spatial repellent is controlled by a single locus bi-allelic autosomal gene, with the deflection allele being completely dominant to the non-deflection allele. (3) Deflected vectors are assumed not to come into contact with the insecticide used in association with the ESR; therefore, mosquitoes that have phenotypes which combine deflection and resistance will not experience any of the fitness benefits associated with resistance if the ESR and insecticide are always present together. (4) The resistance and deflection loci are not linked and re-assort randomly. (5) The genotypes determining adult resistance and deflection phenotypes do not affect the probability of juvenile survival from egg to adult. (6) Mating is random and females mate once, as newly emerged adults, with males in their cohort. (7) Juvenile density dependence means that variation in the absolute number of eggs produced by the adult population does not materially change the rate at which new adults join the population.

Whilst we cannot predict the form that the genetic determinants of resistance, behavioural or otherwise, may take, there are examples of single-locus insecticide-resistance genes, including the knockdown resistance (kdr) alleles that provide resistance to DDT and pyrethroids (*Ndiath et al., 2012*; *Chandre et al., 2000*; *Jones et al., 2012*; *Dabiré et al., 2012*). Whilst the genetic basis of deflection behaviour is unknown and may often be more complex than that of insecticide resistance, it is parsimonious to model this process initially by assuming simple single genes that determine the likelihood of such responses. The impact on the predictions of continuous traits, or, should suitable data become available, of specific more complex genetic assumptions, can be incorporated into future work. The model considers four possible phenotypes, as shown in *Table 1*, with associated genotypes (resistance alleles represented by R, and deflection alleles by D).

The average fitness of offspring into which deflection alleles are inherited, $\overline{F}_D$, is

$$\overline{F}_D = F_S + \frac{[Dr][-R] + [DR]}{[D-]}(F_{RD} - F_S) + \frac{[Dr][-r]}{[D-]}(F_D - F_S) \qquad \text{(Equation 1)}$$

**Table 1.** Phenotype definitions and characteristics.

| Phenotype | Fitness | Resistant | Deflected | Genotypes |
|---|---|---|---|---|
| Susceptible | $F_S$ | No | No | rr\dd |
| Resistant | $F_R$ | Yes | No | Rr\dd RR\dd |
| Deflected and not resistant | $F_D$ | No | Yes | rr\Dd rr\DD |
| Deflected and resistant | $F_{RD}$ | Yes | Yes | Rr\Dd Rr\DD RR\Dd RR\DD |

With $[dr]$, $[dR]$, $[Dr]$, $[DR]$, $[d-]$ and $[D-]$ representing, in the zygote genotypes for the population at a given time point, the proportion of alleles at the deflection locus which are non-deflection alleles paired with susceptible alleles, non-deflection alleles paired with resistant alleles, deflection alleles paired with susceptible alleles, deflection alleles paired with resistant alleles, non-deflection alleles paired with any resistance allele, and deflection alleles paired with any resistance allele, respectively, assuming the same proportions in gametes of mating males and newly emerged females.

The average fitness of offspring into which non-deflection alleles are inherited, $\overline{F}_d$, is

$$\overline{F}_d = F_S + [dR]\left(1 + \frac{[dr]}{[d-]}\right)(F_R - F_S) + \frac{[dr][Dr]}{[d-]}(F_D - F_S) + \left([DR] + \frac{[dR][Dr]}{[d-]}\right)(F_{RD} - F_S) \qquad \text{(Equation 2)}$$

In order for the proportion of deflection alleles in the population to increase, we need the average fitness of the offspring into which deflection alleles are inherited to be greater than the average fitness of the offspring into which non-deflection alleles are inherited. This is true when the following inequality applies:

$$\overline{F}_D > \overline{F}_d \leftrightarrow \frac{[dr]^2}{[d-]}(F_D - F_S) > [dR]\left(1 + \frac{[dr]}{[d-]}\right)(F_R - F_D)$$
$$+ \left([DR]\left(1 - \frac{1}{[D-]}\right) + [Dr]\left(\frac{[dR]}{[d-]} - \frac{[-R]}{[D-]}\right)\right)(F_{RD} - F_D) \qquad \text{(Expression 1)}$$

The equivalent expression for spread of the resistance allele is:

$$\overline{F}_R > \overline{F}_r \leftrightarrow [rd](F_R - F_S) + [rD](F_R - F_D) + \left([rD] + \frac{[RD][-d]}{[R-]} - \frac{[Rd][rD]}{[r-]}\right)(F_{RD} - F_R)$$
$$> \frac{[rd][rD]}{[r-]}(F_D - F_S) \qquad \text{(Expression 2)}$$

See Appendix 2 for derivation of *Equations 1 and 2* and *Expressions 1 and 2*.

We assume that ESR is only relevant where the fitness cost of being susceptible to an IRS insecticide is greater than the fitness cost of being deflected by an ESR, so $F_D > F_S$, requiring that the mortality associated with a mosquito that has a susceptible phenotype entering an insecticide-treated property is greater than that associated with a mosquito being deflected from a property. From *Expression 1*, it can be seen that the spread of the deflection allele in the vector population will be favoured by maximising the fitness difference between susceptible and deflected phenotypes (increasing the value of the left-hand side of the expression), and by maximising the fitness of deflected relative to resistant phenotypes (reducing the value of the right-hand side of the expression). Avoiding the use of ESR in properties without insecticide, minimising $Y_3$ (*Table 2*), and using ESR in all insecticide-treated properties, minimising $Y_2$, improves the survival of deflected phenotypes (given $B<I$) without affecting the survival probability of susceptible or resistant mosquitoes,

**Table 2.** Feeding related survival probabilities.

| | | Feeding-related survival probabilities | | | | |
|---|---|---|---|---|---|---|
| Proportion of properties | | $Y_1$ | $Y_2$ | $Y_3$ | $Y_4$ | |
| Phenotype | Baseline fitness adjustment | Untreated property | Insecticide only | ESR only | Insecticide and ESR | Average survival |
| Susceptible | | $U$ | $U-I$ | $U$ | $U-I$ | $U - I(Y_2 + Y_4)$ |
| Resistant | $COR_1$ | $U$ | $U$ | $U$ | $U$ | $U$ |
| Resistant and deflected | $COR_2$ | $U$ | $U$ | $U-B$ | $U-B$ | $U - B(Y_3 + Y_4)$ |
| Deflected | | $U$ | $U-I$ | $U-B$ | $U-B$ | $U - IY_2 - B(Y_3 + Y_4)$ |

$U$=no-treatment survival, $I$=survival reduction caused by insecticide in susceptible mosquitoes, $B$=survival reduction caused by deflection from protected building, $Y_i$=proportion of properties in each treatment category, $COR_1$=fitness cost of resistance experienced by resistant non-deflected phenotypes, $COR_2$=fitness cost of resistance experienced by resistant deflected phenotypes.

enhancing the desired fitness relationships and hence favouring the spread of deflection alleles in the vector population and the initial establishment of a new evolved spatial repellent.

From *Expression 1*, it can be seen that the spread of deflection alleles is dependent not only upon the relative fitness values of the different phenotypes but also upon there being sufficiently low initial levels of resistance alleles in the population. Since the genotype proportions will change over time, this is a dynamic relationship. In order for the $D$ allele to spread at all, initial allele proportions and fitness relationships must comply with the inequality in *Expression 1*, but the spread of the resistance allele over time may eventually reverse the relationship, so that the proportion of non-deflection alleles will begin to increase instead. The rate of spread of the resistance allele will also determine whether the deflection allele will spread and be sustained in the population. From *Expression 2*, which shows the conditions necessary for the resistance allele to spread, it can be seen that the fitness differential between resistant and susceptible phenotypes and that between resistant and deflected-resistant phenotypes, by helping to determine whether the resistance alleles spread, are also determinants of whether the deflection allele, and hence deflected phenotypes, will spread and be maintained in the population. The spread of the deflection allele when resistance alleles are present in the population is also critically determined by the initial proportion of deflection alleles at the deflection locus and by the proportions of deflection and non-deflection alleles that are paired with resistance alleles. These interactions are explored in *Figures 1–4*.

In the absence of resistance, *Expression 1* reduces to $\overline{F}_D > \overline{F}_d \leftrightarrow [d-](F_D - F_S) > 0$, so deflection would be expected to spread to fixation provided that deflected phenotypes have greater fitness than non-deflected phenotypes. If deflection reaches fixation, then *Expression 2* reduces to $\overline{F}_R > \overline{F}_r \leftrightarrow [rD](F_{RD} - F_D) > 0$, and resistance will only spread if resistant-deflected phenotypes are fitter than deflected phenotypes. In this instance, the strategy of using ESR in all insecticide-treated properties would prevent further spread of resistance because deflected mosquitoes would not enter any insecticide-treated properties and would experience no benefits from resistance. hence the fitness of resistant deflected phenotypes will be the same as or lower than that of non-resistant deflected phenotypes (depending upon cost of resistance), giving $F_{RD} \leq F_D$.

## Results

We carried out a numerical analysis using the model to explore the establishment over time of a new ESR and the associated population-level changes in infectious bite rate. Assumed baseline parameter values are: (i) time from egg laying to adult emergence equivalent to the length of three gonotrophic cycles, (ii) probability per feed that a non-deflected mosquito will acquire a *Plasmodium* infection is 4%, (iii) probability per feed that an infectious mosquito gives an infectious bite on a human host is 80%, (iv) time to infectiousness of *Plasmodium* infection in the vector is approximately equivalent to the length of three gonotrophic cycles. Where not stated otherwise, we use 20% survival of susceptible phenotypes per cycle, 60% survival of resistant phenotypes per cycle, 45% survival of deflected phenotypes per cycle, 0.5% initial proportion of resistance alleles and 25% initial proportion of deflection alleles.

The first question considered is whether, and under what circumstances, selection could generate an effective spatial repellent from a substance that initially repelled only a part of the population. Consistent with *Expression 1,* we found that the spread of a deflection allele through the population depended on the initial proportions of deflection and resistance alleles in the population and the fitness differentials between susceptible, resistant, deflected and resistant deflected phenotypes.

As illustrated in *Figure 1*, for some combinations of fitness values and initial allele proportions, the deflection allele spreads rapidly to near-fixation, and remains consistently at that level for at least 300 cycles (panels labelled '*a*'). In other cases, the deflection allele spreads initially, but falls away within 300 cycles as the resistance allele spreads (panels labelled '*b*'), and in some cases the deflection allele shows only minimal spread before being lost as resistance spreads (panels labelled '*c*'). For cases like that in '*b*', where deflection spreads initially but then falls away, we considered the outcome if the insecticide used is swapped for an alternative, for which resistance alleles are still relatively rare, whilst the deflection allele is close to its peak prevalence. In some cases this allows a 'ratchet' effect, whereby the deflection allele is able to spread and reach sustained high levels.

There is presently little or no direct information available about the fitness costs of a switch to outdoor biting. Furthermore, this would be expected to vary with mosquito species, the degree of

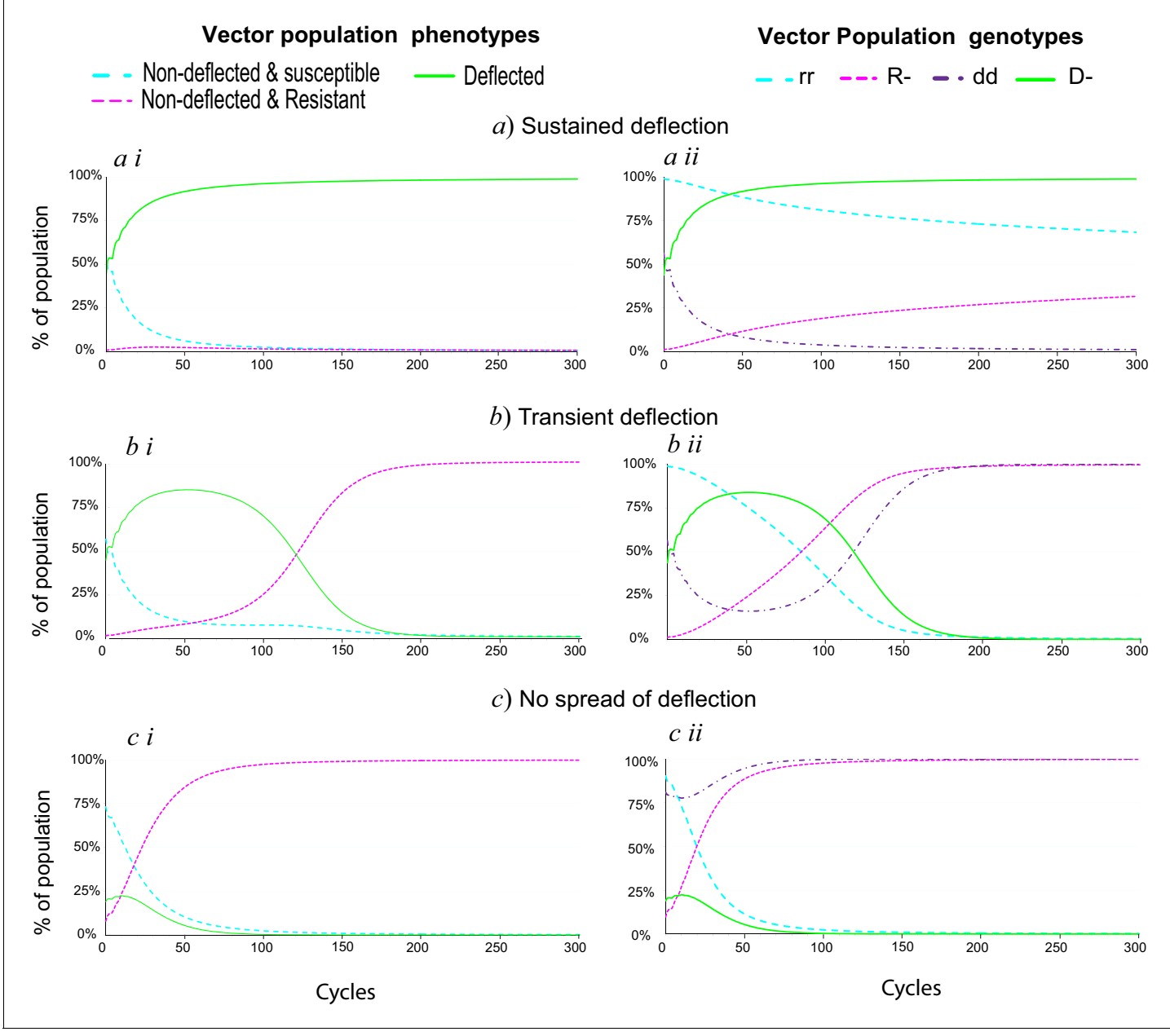

**Figure 1.** Spread of deflection in a population over time. (i) Phenotype and (ii) genotype proportions over time for a population subject to insecticide interventions applied in combination with an ESR. Illustrating (a) long-term establishment of ESR, (b) transient establishment of ESR, and (c) failure to establish ESR. The parameter values used to generate the plots in panels (a) are: 20% per cycle survival of susceptible phenotypes, 60% per cycle survival of resistant phenotypes 45% per cycle survival of deflected phenotypes, 0.5% initial proportion of resistance alleles and 25% initial proportion of deflection alleles. For the panels in (b) and (c) , the per cycle survival of deflected phenotypes is reduced to 40%. For panel (c), other parameters are also amended to 30% per cycle survival for susceptible phenotypes, 5% initial prevalence of resistance alleles and 10% initial prevalence of deflection alleles.

anthropophilly, the type and accessibility of outdoor hosts, and various other factors. The initial proportion of deflection and resistance alleles in the population will depend upon the choice of insecticide and ESR, but will also be expected to vary between specific populations. We therefore carried out analyses for a range of parameter values, the results of which are summarized in *Figure 2*.

A comparison of panels (i) to (iv) in *Figure 2* shows that the fitness values for all phenotypes, and the initial prevalence of deflection and resistance alleles all affect the potential for deflection alleles

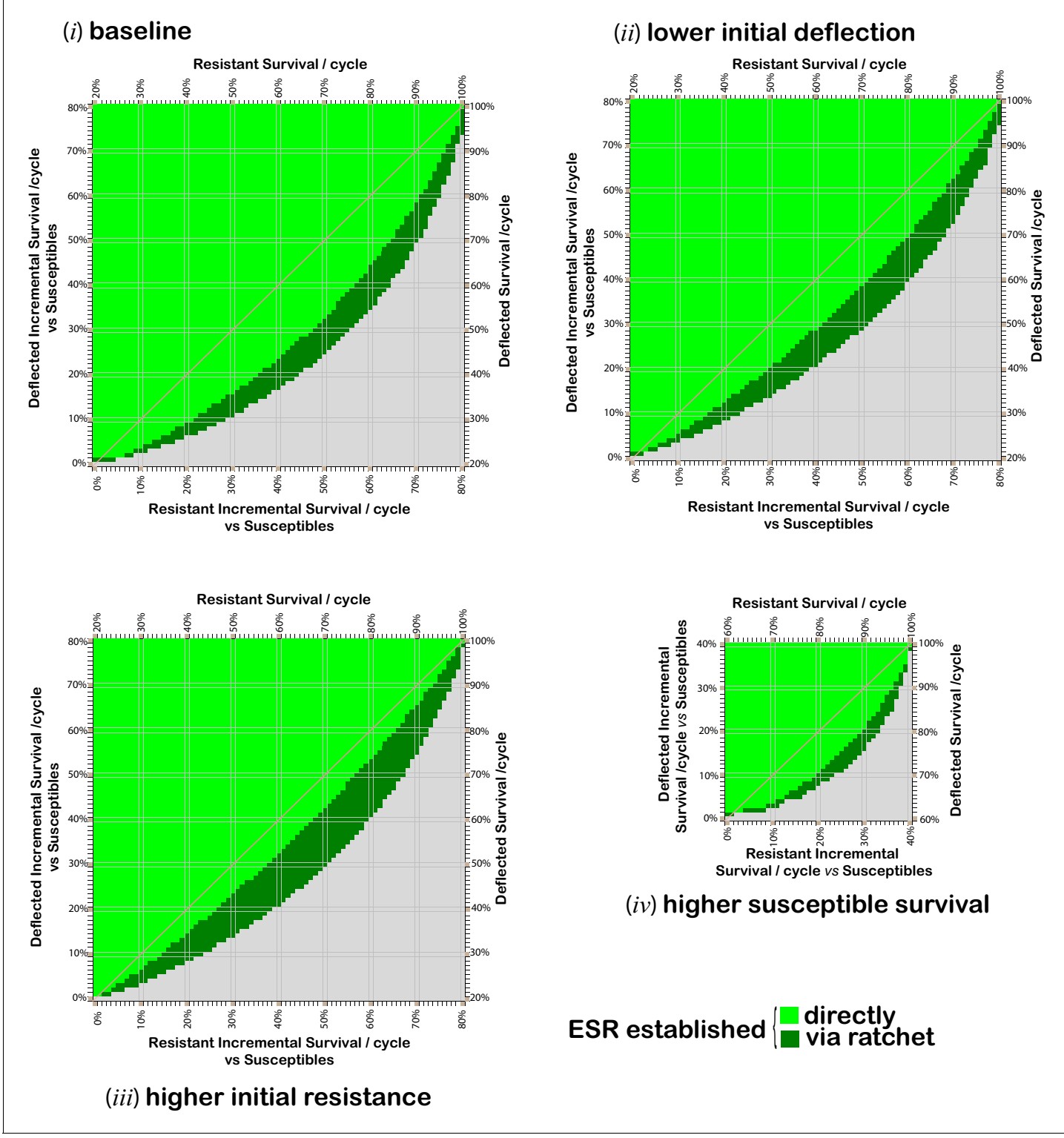

**Figure 2.** Combinations of per cycle survival values for deflected and non-deflected resistant phenotypes, which support the spread and maintenance of deflected phenotypes in the population. Grid plots indicating which combinations of resistant (*x*-axis) and deflected (*y*-axis) phenotype per-cycle survival values (in 1% increments) give rise to a population comprising at least 80% deflected phenotypes after 300 modeled time periods. When this is achieved directly, the applicable square is bright green. Dark green squares indicate combinations for which the required outcome can be achieved via a 'ratchet' where the initial paired insecticide is swapped once for a new insecticide, with allele proportions at the time of the swap assumed to be 0.5% resistance alleles and the maximum percentage of deflection alleles achieved whilst using the first insecticide. Results are calculated for 1% increments

*Figure 2 continued on next page*

*Figure 2 continued*

in each survival value. Gridlines and diagonals are to aid visual location of results on the grid. The baseline parameters (panel i) are: 20% per cyclesurvival of susceptible phenotypes; resistant deflected phenotypes have the same survival probability as non-resistant deflected phenotypes; 0.5% initial prevalence of resistance alleles; and 25% initial prevalence of deflection alleles. Parameter values for panels (ii) to (iv) differ from the baseline values as follows: panel (ii) 10% initial prevalence of deflection alleles; panel (iii) 2% initial prevalence of resistance alleles; and panel (iv) 60% per cycle survival of susceptibles .

The following figure supplement is available for figure 2:

**Figure supplement 1.** Effect of incomplete deflection on fitness combinations which support the spread and maintenance of deflection.

to spread and be maintained in the population, consistent with the relationships shown in *Expression 1*. For example, with the baseline parameter values, in a context in which insecticide-resistant phenotypes have average per-cycle survival of 60% and deflected phenotypes have per-cycle survival of 45%, an ESR introduced at a time when the prevalence of resistance and deflection alleles are 0.5% and 25% will become established, with the deflection allele spreading and deflected phenotypes comprising more than 80% of the population 300 cycles after introduction (panel i), as shown

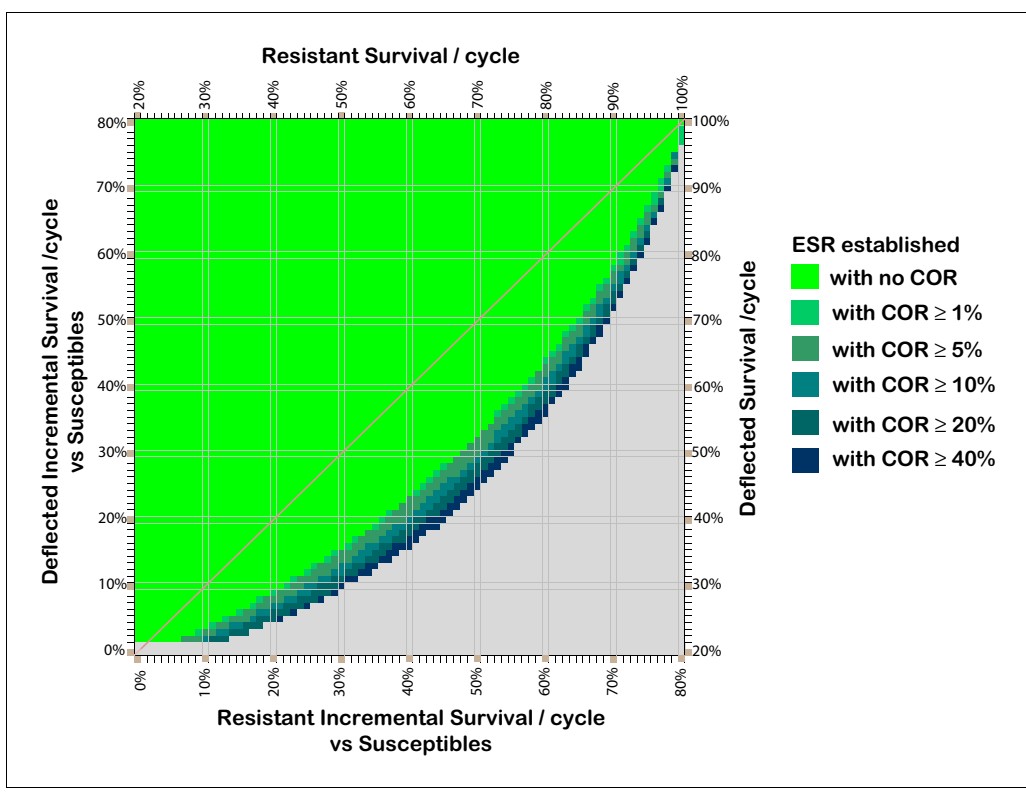

**Figure 3.** Effect of cost of resistance on resistant and deflected per-cycle survival combinations which support the spread and maintenance of deflected phenotypes in the population. Combinations of phenotype survival values which can result in more than 80% of the population having deflected phenotypes after 300 cycles, assuming various costs of resistance. Colours indicate the lowest cost of resistance (COR) incurred by deflected resistant phenotypes which achieves the threshold 80% deflection phenotypes in the population after 300 cycles (without assuming any ratchet). COR here represents the reduction in per cycle survival of deflected resistant phenotypes arising as result of having a resistant phenotype, so the resistant deflected phenotype has per-cycle survival equal to that for the deflected phenotype (*y*-axis) less the applicable COR. The survival values shown for the resistant phenotype (*x*-axis) are those after taking account of any cost of resistance that affects non-deflected resistant phenotypes. The baseline parameters are: 20% per cycle survival of susceptible phenotypes; 0.5% initial prevalence of resistance alleles; and 25% initial prevalence of deflection alleles.

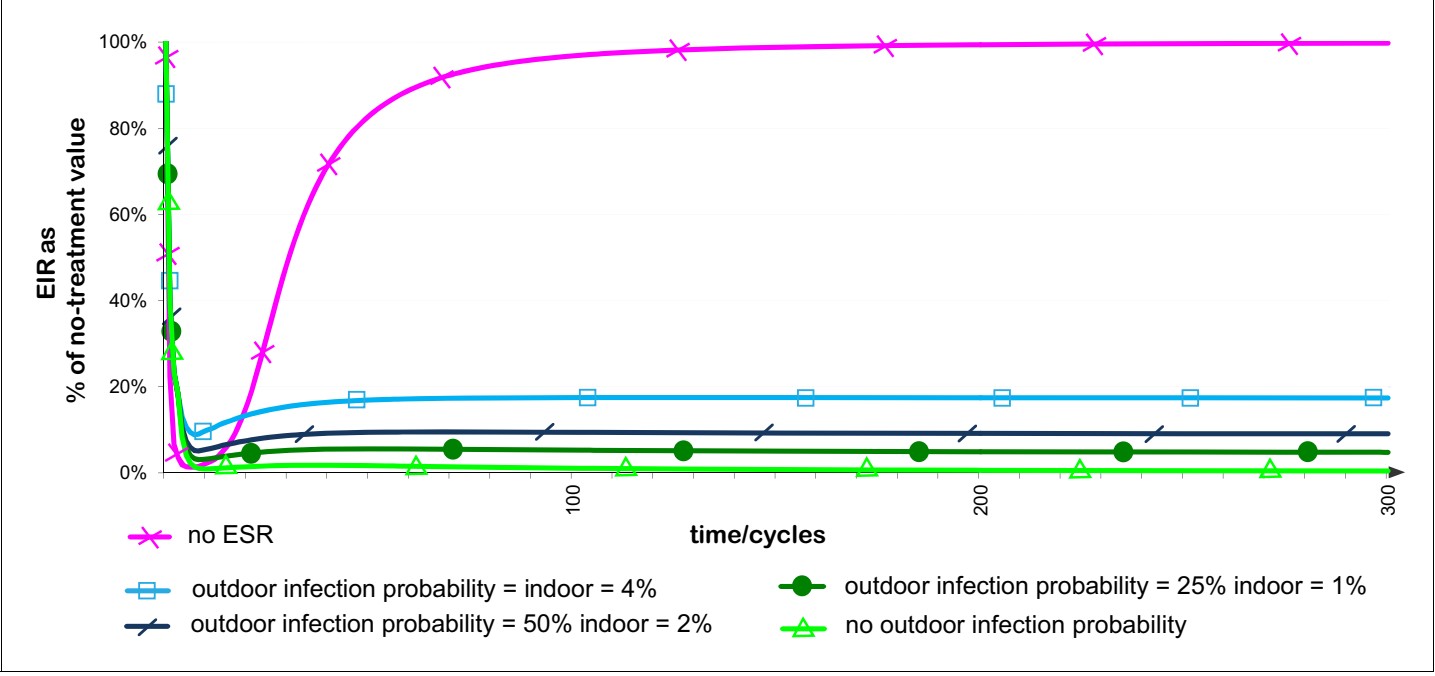

**Figure 4.** Effect of evolved spatial repellent on infectious bites from a vector population over time. The plots represent the infectious bites from the vector population per unit of time as a proportion of that with no intervention, assuming use of insecticide with and without ESR and probabilities per feed that ESR-deflected mosquitoes will become infected with *Plasmodium* of 4%, 2%, 1% or 0%. Plots otherwise use baseline parameter values. Probability per feed that non-deflected mosquito acquires *Plasmodium* infection is assumed to be 4%. If the size of the human population is assumed to be the same for all treatments and time periods, this equates to the entomological inoculation rate (EIR) as a percentage of the EIR with no intervention.

by the light green square for 45% deflected survival with 60% resistant survival. However, if the initial prevalence of deflection alleles is only 10%, then with the same per-cycle survival rates, deflection alleles spread but are not sustained. Deflection can, however, be established by replacement of the initial insecticide whilst deflection alleles are at their maximum prevalence (panel *ii*), as shown by the dark green square for 45% deflected survival with 60% resistant survival.

From comparison of *Figure 2* panel (*i*) and *Figure 3*, it can be seen that if a cost of resistance affects resistant deflected phenotypes this serves to increase the range of deflection and resistance fitness combinations for which deflection alleles can spread and be maintained.

Once the ESR is established, the impact of the ESR–insecticide combination treatment on the transmission of *Plasmodium* will critically depend on the reduced vectorial capacity arising from the exclusion of mosquitoes from treated properties. This will have two components. Reduced survival of deflected mosquitoes will reduce the population of adult mosquitoes, the probability of infected mosquitoes surviving to give an infectious bite, and the number of bites that an infectious mosquito will survive to give. In addition, anything which reduces the probability of transmission from human host to a feeding mosquito, or from a feeding mosquito to a human host, will enhance the reduction in transmission arising from deflection away from human dwellings. The level of transmission of *Plasmodium* to or from mosquitoes that have transitioned to outdoor feeding is not yet well-explored. In calculating the levels of infectious bites that correspond to given levels of deflected mosquitoes in the population, we therefore represent all the possible sources of reduced transmission by deflected vectors as a range of possible parameter values for their probability per feed of acquiring a *Plasmodium* infection, assuming probabilities of 4%, 2%, 1% and 0% (100%, 50%, 25% or 0% of the value assumed in the absence of deflection). From *Figure 4*, it can be seen that for a context in which transmission is as efficient for deflected outdoor-feeding vectors as for vectors exposed to no intervention, the initial reduction in infectious bites achieved when using an ESR (blue line with squares) is comparable with, but not quite equal to, the reduction in infectious bites achievable using

an insecticide alone (pink line with crosses). However, the reduction in bites achieved using the ESR is maintained at a high level in the long term, while resistance rapidly eliminates the effectiveness of the unpaired insecticide. When assuming some reduction in transmission for deflected mosquitoes (lines with slash, circle and triangular markers), this trade-off between immediate and long-term benefits is reduced, with the ESR offering an initial reduction in infectious bites very similar to that achieved initially with insecticide alone, with the benefit again maintained or improved over the long term.

Although the results summarized in *Figures 1*, *2* and *4* assume that resistant deflected phenotypes have the same fitness as susceptible deflected phenotypes, consistent with a context in which deflected phenotypes never enter a property treated with insecticide, and hence never experience any fitness benefit from resistance, note that this is not a requirement for successful establishment of an ESR, as illustrated in *Figure 2—figure supplement 1*.

## Discussion

The use of DDT for indoor residual spray (IRS) programs proved an outstanding success in the history of public health campaigns against malaria. Its withdrawal in the light of environmental concerns reversed successes that approached elimination in some regions (*Roberts, 2010*; *Curtis, 2002*; *Roberts et al., 1997*). By the time of its withdrawal, resistance to the toxic effects of DDT was already widely observed, but there is empirical evidence to suggest that it may nonetheless have maintained efficacy as a transmission-reduction agent through its action as an effective spatial repellent (*Roberts et al., 2000*; *Roberts and Alecrim, 1991*). In part inspired by this, the deflection of malaria-vector mosquitoes from indoor feeding at night to outdoor feeding is being actively investigated as a means to reduce malaria transmission (*Achee et al., 2012*; *University of Notre Dame, 2016*). Here, we show that combining a spatial repellent which initially repels only a small proportion of a target vector population with indoor residual spraying of a high-toxicity insecticide can serve both to create a highly effective spatial repellent and to protect the companion insecticide from the rapid evolution of direct resistance to its toxic effects.

Our analysis provides the initial theoretical framework to spur empirical testing of this concept. There is, however, already a body of empirical evidence consistent with our proposal that selection can act to increase the efficacy of a repellent paired with a toxic insecticide. Our work predicts that, given heritable behavioural traits in the exposed population, toxic substances encountered with sufficient frequency and with sufficient volatility to be detectable at a distance would be observed to show repellent action against exposed populations. In fact, many substances investigated because of their repellent properties are also toxic on contact with target species. These include DEET, probably the most effective and well-known of the available repellents, as well as many of the naturally repellent plant-based volatiles currently in use or under investigation for personal protection (*Xue et al., 2003*; *Licciardi et al., 2006*). In fact, experiments that have combined repellents with insecticides on bed nets have often found that the benefits arise from the additive or synergistic toxicity of the repellents (*Pennetier et al., 2007*; *Faulde et al., 2010*; *N'guessan et al., 2008*) to an equivalent or greater degree than through their localized repellent effects. A comparison of the behavioural and insecticidal effects of three public-health insecticides provides more direct evidence of toxicity giving rise to repellency. The most toxic of the three, carbosulfan, also demonstrated high spatial repellence in a population of susceptible mosquitoes but not in a resistant population (*Malima et al., 2009*), an observation wholly consistent with our theoretical predictions.

The two elements defining the public-health benefits of an ESR used to deflect vectors away from human dwellings are: (1) the proportion of the mosquito population affected by the ESR; and (2) the reduction in infectious bites resulting from deflection. The former is dependent on the spread of the deflection allele in the population. This is determined by the relative fitness of susceptible, resistant, deflected and resistant-plus-deflected phenotypes, and by the initial prevalence of deflection and resistance alleles when the combined ESR-insecticide intervention is introduced. Choices about how the ESR and its associated insecticide are selected and deployed can influence or determine these values. The potential for establishing an ESR is maximized by lower fitness values for susceptibles and by higher fitness values for deflected phenotypes, indicating that control programs should target high coverage with high-efficacy insecticides to minimize the fitness of non-deflected susceptibles. Critically, all treated properties should ideally be treated with both an insecticide and ESR,

rather than with either alone, in order to maximize selection for deflection and to minimize selection for physiological resistance. Minimising the entry of deflected phenotypes to insecticide-treated properties avoids exposing deflected phenotypes to insecticide and hence reducing the average fitness of deflected phenotypes relative to that of resistant phenotypes. Minimising the use of ESR on properties without insecticide coverage, particularly during the establishment stage of an ESR, avoids generating a fitness cost for deflected mosquitoes but not for non-deflected susceptible mosquitoes which can enter and feed. The use of a well-established ESR in a small number of properties for which insecticide use is for some reason impossible may, however, comprise one of ESR's potential benefits. A high initial proportion of deflection and a low proportion of resistance alleles in the treated population also supports the spread of deflection alleles and the establishment of an ESR. Although these values cannot be directly controlled, desirable values may be targeted through careful choice of insecticide and ESR for each vector population. The candidate ESR can be chosen to target an initial threshold level of deflection genotype in the population. The initial companion insecticide can be chosen to target low initial levels of physiological resistance in the target population, and could exploit high-toxicity insecticides that would normally show a very rapid loss to resistance, serving both to maximize selection for deflected phenotypes and to give maximum immediate transmission reduction benefits.

An ESR needs to be partnered with a suitable IRS insecticide, which should have low contact repellence to maximize the mortality produced in non-deflected mosquitoes. Existing and new chemical insecticides with low contact repellence and high toxicity, such as bendiocarb (*Evans, 1993*), are therefore suitable as potential partner insecticides for an ESR program, potentially transforming them from short-term solutions to sustainable tools. Furthermore fungal biopesticides being developed for vector control potentially offer an ideal partnership with a spatial repellency treatment, whether DDT or a novel chemical, as they appear to have no contact repellency and have inherent resistance management benefits (*Thomas and Read, 2007*; *Mnyone et al., 2010*). These novel biopesticides have previously been proposed as 'evolution proof' late-life-acting insecticides (*Read et al., 2009*; *Thomas and Read, 2007*), offering respite from the treadmill of insecticide loss to resistance. For late-life-acting insecticides, relatively low-virulence fungal strains are ideal (*Lynch et al., 2012*), but much work on these organisms has focussed on generating high-virulence strains (*Fang et al., 2012*). Given their low or absent contact repellence, such high-virulence strains would provide ideal candidates for combination with an evolved spatial repellent, provided action can be taken before resistance becomes established in the target mosquito populations.

Our focus is on repellence that prevents indoor-feeding malaria vectors from entering properties, rather than on more localized repellence away from bed nets once vectors have entered properties. The excito-repellent properties of some of the pyrethroid insecticides currently used on long-lasting insecticide-treated nets (LLINs) reduce the mortality they generate by pushing vectors away before they acquire a lethal dose (*Grieco et al., 2007*; *Tananchai et al., 2012*). Such reduced mortality would diminish their likely efficacy as partner insecticides for ESRs. They could nonetheless enhance the establishment of an ESR when deployed in combination with suitable IRS, by reducing the relative fitness both of non-deflected susceptible mosquitoes and of mosquitoes resistant to the IRS insecticide. Such insecticides would also benefit from the resistance protection provided by an established ESR. As one of the key public-health tools currently deployed, and one already showing signs of succumbing to resistance (*N'Guessan et al., 2007*; *Ochomo et al., 2014*; *Chandre et al., 1999*), protecting the efficacy of these compounds could be hugely beneficial. Further, in the search for replacements for pyrethroids on bednets, alternative actives with low contact repellence could make excellent partner insecticides for an ESR and again benefit from protection against a rapid loss to resistance. If the partner insecticide is deployed via bednets then, as for IRS, the ESR can be deployed separately, allowing it to be applied, refreshed, removed or replaced without requiring any changes to the manufacture or maintenance of the LLINs.

The fitness cost of deflection away from indoor human hosts will vary according to vector species and degree of anthropophilly. Clearly for species that are readily zoophagous, such as *Anopheles arabiensis*, the fitness cost of diverting to a livestock host would be relatively low, provided that the alternative host is present and accessible. For more anthropophilic species, such as *Anopheles gambiae*, the fitness cost of deflection must be higher, and may be expected to vary between local populations depending on the feeding alternatives adopted. In assessing where ESRs offer high

potential to contribute to public health campaigns, therefore, consideration should be given to the detail of the species mix in the local population, and the availability of alternative hosts. Immigration into the population will slow the spread of deflection, and so the size of a treated area is also a likely determinant of success since a large enough treatment area can minimize immigration by effectively including a whole breeding population within the treatment area. Equally, an isolated settlement may provide a closed vector population over a relatively small treatment area.

The separation of repellence and toxicity provides additional benefits. The companion insecticide can be changed whilst maintaining use of an established ESR. In contexts where an ESR can only be established on a transient basis, for instance because resistance alleles are already relatively common in the population when the ESR is introduced or because the population is subject to sustained immigration of non-deflected phenotypes, the ESR may still be established using a 'ratchet' approach in which the insecticide used in combination with the ESR can be changed when deflection phenotypes reach their peak, so that alleles for resistance to the new product will only offer a fitness benefit if paired with the relatively small proportion of non-deflection genotypes remaining in the population, allowing the deflection allele to spread. Where an ESR can become well established in populations in which the fitness of resistant phenotypes is higher than that of deflected phenotypes, unless the deflection allele reaches fixation, insecticide resistance will still spread eventually and deflection will eventually disappear. However, replacement of the partner insecticide at a suitable time will preserve the benefits of the established ESR, and the spread of resistance to the new partner insecticide may be wholly suppressed. This might open public-health opportunities such as the short-term use of a more expensive insecticide to generate ESR protection for the long-term use of a cheaper alternative.

When the conditions for establishment of an ESR are met, then differential transmission for outdoor/early biting *vs* indoor biting is potentially very important for the outcome in terms of reducing infectious bites. Because it is easiest to establish an ESR where deflection has little impact on fitness, if the only mechanism by which deflection to outdoor biting has an effect on infectious bites is through the incremental mortality associated with outdoor biting, then ESRs are most useful in the circumstances where they are hardest to establish. However, many other factors may reduce the probability per feed that a vector will acquire *Plasmodium*, or that, once infectious, it will transmit the parasite to a human host.

There is empirical evidence of reduced *Plasmodium* infection in at least one vector population which has transferred from indoor to outdoor biting (*Ndiath et al., 2014*). Ndiath et al. (*Ndiath et al., 2014*) found no sporozooites in a population of insecticide-susceptible *An. gambiae* pushed to outdoor feeding by the deployment of LLINs in Dielmo, whilst transmission continued at a high level in an indoor-feeding resistant population (*Ndiath et al., 2014*). Whilst this specific example cannot be unambiguously attributed to reduced transmission to/from outdoor feeding vectors rather than, for example, to differential mortality rates, it is reasonable to consider other potential causes of reduced outdoor transmission rates. These might include, for example, increased probabilities of taking feeds from non-human hosts (*Lefèvre et al., 2009*), differential availability of infectious human hosts indoors and outdoors, reduced effectiveness of transmission outside normal biting times (*Gautret and Motard, 1999*; *Mideo et al., 2013*; *O'Donnell et al., 2011*), and increased active host response to early/outdoor feeding attempts. Deflection to outdoor biting could therefore have dramatic effects on transmission that are independent of direct vector mortality effects, and there is potential for easily established ESRs which are as effective as conventional insecticides in reducing transmission, but on a much more sustainable basis.

There is still some controversy about the desirability of deflecting mosquitoes away from indoor biting and the well-established control methods which exploit this behaviour. The spatial repellent concept (whether evolved or conventional) is predicated on the idea that forcing vectors and *Plasmodium* into behavioural options that offer them lower fitness outcomes will inherently provide a new and potentially sustainable means of reducing transmission (*Achee et al., 2012*). However, outdoor biting is commonly viewed as a route to increased vector activity in a context where personal protection and anti-vector measures are hard to action (*Bradley et al., 2012*). In some contexts, for some compounds, the ESR concept may help to generate and sustain repellents that are able to protect against outdoor biting in the vicinity of treated properties, but in some locations it may nonetheless be the case that producing a move to outdoor biting simply does not generate overall transmission reductions equivalent to those achieved by control measures applied directly to indoor-

feeding vectors. Transmission reductions may also change over time; for example, if outdoor biting initially generates low transmission because the accessible outdoor human hosts are all adults who have partial immunity, successful protection of children would eventually generate increasing numbers of susceptible adults who will in turn be exposed to outdoor feeding vectors, eliminating this benefit. It might also be argued that, whether beneficial or detrimental, outdoor biting will inevitably evolve as a behavioural resistance mechanism, and therefore that the use of an ESR is irrelevant. However, we would argue that ESRs offer benefits even in contexts where a transition to outdoor biting is ultimately found to be detrimental. For vectors that preferentially feed indoors, there is an expected fitness cost to a switch to outdoor feeding. Selection should therefore favour the use of a 'cue' to determine host choice, allowing the fitness benefits of indoor feeding to be enjoyed wherever safe, whilst avoiding the costs of entering insecticide-treated properties. Where ESRs are used in such contexts, either they will have no effect at all (i.e. they will do no harm nor good) or at least part of the vector population selected to respond to a 'cue' will be 'cued' by the ESR. This means that, should outdoor feeding prove undesirable at some point, the transition to outdoor feeding will be at least partly reversible since, unlike direct responses to an insecticide or to entering buildings, the response to an ESR will cease to affect vector behaviour if the ESR is withdrawn.

As well as being reversible in its effects, unlike many resistance-management strategies, ESR does not require the withdrawal or reduction of existing control measures in order to be effective. All novel interventions carry potential for harm, however, if they divert limited resources away from more effective control measures. In order to minimize potential harm as well as maximising potential benefits therefore, candidate ESRs should be cheap to obtain, and cheap and easy to deploy in the field. Unlike IRS, there is no imperative to cover interior surfaces with an ESR as they are intended to act at a distance, deployment as a single-spot event should therefore be possible. If this is formatted as a simple physical item, for example a small disc of impregnated paper, it can be deployed without the need for any special equipment. For ease of acceptance, candidate ESRs should have no detectable unpleasant odour for humans. To minimize unintended fitness costs for deflection, candidate substances should not be common in the natural environment of the target vector population. Any carrying material used for ESR distribution should be cheap, readily available, not amenable to other practical use, resistant to degradation and inedible for animals with indoor access.

The evolution of behavioural change in response to pesticides is also a recognized issue for agricultural insecticide use (*Castillo-Chavez et al., 1988*; *Kennedy et al., 1987*; *Jongsma et al., 2010*) and it is interesting to note that in the early 1980s *Gould (1984)* suggested that some insect populations might be expected to evolve behavioural avoidance of agricultural insecticides which were not originally repellent to them, and that in many situations, physiological resistance would evolve more slowly to insecticide formulations with high repellency than to non-repellent formulations. Whereas use in agriculture is commonly highly detrimental to the sustained utility of public-health tools, the parallel potential for ESRs in human health and agriculture may in some contexts allow agricultural use to enhance their public health role. In areas where vectors rest in insecticide-treated crops, using a vector-control spatial repellent on crops in combination with agricultural insecticides may serve to enhance selection for deflection alleles without generating any selection pressure for physiological resistance to public health insecticides (whilst incidentally providing some resistance-management for agricultural pesticides). This might also be a means to apply selection to nectar-feeding males, or to enhance selection for deflection in contexts where there is some existing resistance to all available public health insecticides but not to the agricultural insecticides being used locally.

Work carried out since the widescale use of DDT was terminated has clarified the role of spatial repellence in its contribution to malaria control (*Roberts et al., 2000*). Given its highly effective spatial repellence combined with high toxicity and the ubiquity of its use, it is perhaps interesting to consider whether DDT was always an effective repellent, or whether it actually provides the first empirical example of the ESR concept in action?

Our models suggest that there is clear potential to use evolution to create better control through evolved spatial repellence, and we hope to stimulate empirical work to test both our assumptions and the application of this novel approach. More generally, we demonstrate the importance of taking into account and modeling the evolutionary implications of different methods of insect control and medical interventions more broadly. We provide an example in which the inevitable evolution of the target insects can be used to improve rather than to reduce the effectiveness of the intervention.

When evaluating a new intervention detailed mathematical modeling of the evolutionary outcomes alongside the more commonly considered epidemiological outcomes has considerable potential to improve infectious disease control. We recommend bringing together theoretical and empirical work to explore fully the potential of the ESR concept.

## Acknowledgements

The authors would like to thank Andrew Read for early discussions of the ESR concept. We would also like to thank the editors and reviewers at eLife who were extremely helpful and supportive during the process of bringing the manuscript to publication. They helped to improve the manuscript substantially and they introduced a number of interesting ideas, including the potential for ESR to support repellents for use outdoors.

## Additional information

### Funding

| Funder | Grant reference number | Author |
| --- | --- | --- |
| Biotechnology and Biological Sciences Research Council | BB/L010879/1 | Penelope Anne Lynch Mike Boots |
| NERC Environmental Bioinformatics Centre | NE/J009784/1 | Penelope Anne Lynch Mike Boots |

The funders had no role in study design, data collection and interpretation, or the decision to submit the work for publication.

### Author contributions

PAL, Conception, design and modelling, Drafting or revising the article; MB, Design, Drafting or revising the article

### Author ORCIDs

Penelope Anne Lynch, http://orcid.org/0000-0002-0486-8507

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

## Appendix 1

# Definition of the PM model

**Appendix 1—table 1.** Variable and parameter definitions for the population genetics model.

**Variable and parameter labels**

| Label | Description | Units |
|---|---|---|
| $t$ | Index of model time periods<br>*length of one time period equivalent to duration of one gonotrophic cycle* | Time |
| $h$ | Index of phenotypes<br>1= susceptible non-deflected<br>2 = resistant non-deflected<br>3 = resistant and deflected<br>4 = susceptible deflected | |
| $g$ | Index of genotypes<br>1=rrdd, 2=rRdd, 3=RRdd, 4=rrDd, 5=rRDd,<br>6=RRDd, 7=rrDD, 8=rRDD, 9=RRDD | |
| $D_{w,t}$ | The proportion of male gametes with allele pair $w$ in time period $t$.<br>Pairs are: 1=rd, 2=rD, 3=Rd, 4=RD | Proportion of male gametes |
| $T_t$ | Infectious bites from population in period $t$ as a proportion of baseline (pre-treatment) infectious bites per period | Proportion of baseline bites/time |
| $I_t$ | Females giving infectious bites in period $t$ as a proportion of females in baseline population | Proportion of baseline population |
| $M_{h,t}$ | Females of phenotype $h$ which are infectious with *Plasmodium* in period $t$ as a proportion of baseline population | Proportion of baseline population |
| $c_0$ | Baseline probability that an infectious female in the no-intervention baseline population will give an infectious bite when feeding | Probability |
| $c_h$ | Probability that an infectious female of phenotype $h$ will give an infectious bite when feeding | Probability |
| $m_{h,t}$ | Females newly infected with *Plasmodium* in period $t$ as a proportion of the baseline population | Proportion of baseline population |
| $\beta_0$ | Baseline average probability susceptible individuals survive one model time period (in the absence of any intervention) | Probability |
| $\beta_h$ | Average probability that individuals with phenotype $h$ will survive one time period in an environment including a given intervention | Probability |
| $\gamma$ | Number of model time periods between start of *Plasmodium* infection in mosquito, and point from which infection has matured and mosquito can give infectious bites | Model time periods |
| $b_0$ | Probability that an uninfected mosquito in a baseline population subject to no interventions will acquire a *Plasmodium* infection during a single model time-period | Probability |
| $b_h$ | Probability that an uninfected mosquito with phenotype $h$ will acquire a *Plasmodium* infection during a single time-period | Probability |
| $U_{h,t}$ | Females with phenotype $h$ which are not infected with *Plasmodium* at start of time period $t$, as a proportion of the baseline population | Proportion of baseline population |
| $J_{h,t}$ | New females of phenotype $h$ in period $t$, as a proportion of the baseline population | Proportion of baseline population |
| $B_{g,t}$ | Proportion of new adults in time period $t$ with genotype $g$ | Proportion of new adults |
| $\tau_{1,t}$ | New (female) adults in period $t$ as proportion of baseline population | Proportion of baseline population |
| $\tau_{2,t}$ | Adults surviving from preceding period to start of period $t$ as proportion of baseline population | Proportion of baseline ppopulation |
| $\tau_{3,t}$ | New (female) adults in period $t$ as a proportion of total (female) population in period $t$ | Proportion of current population |

*Appendix 1—table 1 continued on next page*

*Appendix 1—table 1 continued*

**Variable and parameter labels**

| Label | Description | Units |
|---|---|---|
| $\tau_{4,t}$ | Adults surviving from preceding period to start of period $t$ as proportion of population in period $t$ | Proportion of current population |
| $\tau_{5,t}$ | Population in period $t$ as proportion of baseline population | Proportion of baseline population |
| $S_g$ | Probability a female with genotype $g$ will survive one model time period | Probability |
| $E_{g,t}$ | Proportion of all eggs laid in time period $t$ which have genotype $g$ | Proportion of eggs laid in period |
| $K_{g,t}$ | Proportion of adult females surviving at start of time period $t$ which have genotype $g$ | Proportion of adult females |
| $\varpi$ | Number of time periods between egg-laying and emergence of new adults | Model time periods |
| $W_{g,t}$ | The proportion of all females (including new adult females) which have genotype $g$ at start of time period $t$ | Proportion of adult females |
| $\psi_t$ | Proportion of the population alive at start of time period $t-1$ which survive to start of time period $t$ | Proportion of population |
| $\theta_{g,t}$ | Number of eggs with genotype $g$ laid in period $t$, relative to a normalized number of eggs assumed per mosquito per successful lay | Relative number of eggs |
| $\delta_t$ | Total number of eggs laid in period $t$, relative to a normalized number of eggs assumed per mosquito per successful lay | Relative number of eggs |
| $Z_{g,j,t}$ | The proportion of eggs produced by females which are new adults in time period $t$ and have genotype $j$, which are genotype $g$ | Proportion of eggs |
| $A_{g,t}$ | Average number of eggs laid during time period $t$ by females with genotype $g$ alive at start of time period $t$ | Normalized number of eggs |
| $L$ | Normalized number of eggs assumed per mosquito per successful lay | Eggs/mosquito/lay |
| $X_t$ | Newly mated females in time period $t$ as a proportion of the baseline population | Proportion of baseline population |
| $G_{1,t}$ | Proportion of alleles at the resistance locus which are resistance alleles in model time period $t$ | Proportion of alleles |
| $G_{2,t}$ | Proportion of alleles at the deflection locus which are deflection alleles in model time period $t$ | Proportion of alleles |
| $P_{1,t}$ | Proportion of population with resistant phenotypes in model time period $t$ | Proportion of current population |
| $P_{2,t}$ | Proportion of population with deflected phenotypes in model time period $t$ | Proportion of current population |
| $P_{3,t}$ | Proportion of population with susceptible non-deflected phenotypes in model time period $t$ | Proportion of current population |
| $V_{y,t}$ | Proportion of population with genotype $y$ in model time period $t$ genotypes; 1=rr, 2=rR or RR, 3=dd, 4=dD or DD | Proportion of current population |

The model tracks the population of adult females and assumes a 'baseline population' of stable size (new adult recruitment rate is constant and equal to adult mortality rate) in the absence of any insecticide or ESR interventions. It is assumed that larval habitats are over-subscribed, and variations in the adult population do not materially change the number of juveniles reaching maturity and joining the adult population in each time period. Females are assumed to mate once, as newly emerged adults, mating with males having the same genotype proportions as newly emerged adult females.

$t$ represents the index of model time periods, with $t=0$ representing the periods prior to application of an intervention when the population is the baseline population.

$S_g$, the probability that a female with genotype $g$ will survive one time period, is the primary input value driving the spread of different alleles in the popgen model. Each genotype is

associated with one of the four possible phenotypes. Each phenotype has an associated per period survival probability.

Infectious bites from population in period $t$ as a proportion of baseline infectious bites, $T_t = \dfrac{I_t}{I_0}$.

Females giving infectious bites in period $t$ as a proportion of females in the baseline population, $I_t = \sum\limits_{h=1}^{4} M_{h,t} c_h$.

The females of genotype $h$ that are newly infected with *Plasmodium* in period $t$, as a proportion of the baseline population, $m_{h,t}$, is calculated as $m_{h,0} = U_{h,0}\beta_0 b_0$ and $m_{h,t} = U_{h,t}\beta_h b_h$ with $t>0$.

$M_{h,t}$ females of phenotype $h$ that are infectious with *Plasmodium* in period $t$ as a proportion of baseline population, calculated for the baseline population as the total proportion of newly infected females in each period, for all periods up to $\gamma$ periods (number of periods between *Plasmodium* infection and infectiousness) ago, multiplied by the probability of surviving to $\gamma$ periods ago, multiplied by the probability of surviving $\gamma$ periods, $M_{h,0} = \dfrac{m_{h,0}}{1 - \beta_0}\beta_0{}^{\gamma}$.

After baseline period, $M_{h,t}$ is calculated for each phenotype as surviving infectious females from the previous period, plus survivors from newly infected females with that phenotype from $\gamma$ periods ago. $M_{h,t} = M_{h,(t-1)}\beta_h + m_{h,(t-min(t,\gamma))}\beta_h{}^{\gamma} \qquad t>0$

Females with phenotype $h$ that are not infected with *Plasmodium* at start of time period $t$, as a proportion of the baseline population.

For the baseline population, $U_{h,0}$, the non-infected females at the start of each time period are calculated for each phenotype as the sum of the surviving, uninfected new females from all past time periods, assuming a constant value for new females of each phenotype in each period, $U_{h,0} = \dfrac{J_{h,0}}{1 - \beta_0(1 - b_0)}$. Post-introduction of the intervention, the non-infected females for each genotype at the start of each time period are calculated as the surviving non-infected females from the previous period plus new females of that genotype, giving $U_{h,t} = U_{h,t-1}\beta_h(1 - b_h) + J_{h,t}$ with $t>0$.

$J_{h,t}$, new females of phenotype $h$ in period $t$, expressed as a proportion of the baseline population, is calculated as the proportion of the new adults in period $t$ that have genotypes corresponding to phenotype $h$.

New susceptible females in time period $t$ as a proportion of the baseline population,

$$J_{1,t} = B_{1,t}\tau_{1,t}.$$

New deflected females in time period $t$ as a proportion of the baseline population,

$$J_{2,t} = \left(B_{4,t} + B_{7,t}\right)\tau_{1,t}.$$

New resistant females in time period $t$ as a proportion of the baseline population,

$$J_{3,t} = \left(B_{2,t} + B_{3,t}\right)\tau_{1,t}.$$

New deflected-resistant females in time period $t$ as a proportion of the baseline population,

$$J_{4,t} = \left(B_{5,t} + B_{6,t} + B_{8,t} + B_{9,t}\right)\tau_{1,t}.$$

For this analysis, it is assumed that larval habitats are over-subscribed and that the number of new adults recruited to the population in each time period remains constant with and without interventions. In all time periods therefore, the number of new (female) adults joining the population is the same as for the baseline population. Since the baseline population is assumed to be constant, the proportion of the population comprising new adults in each time period in the baseline population can be calculated as $\tau_{1,t} = 1 - \beta_0$.

$\tau_{2,t}$, adult females surviving from the preceding period to the start of period $t$ as a proportion of the baseline population, is calculated as $\tau_{2,0} = 1 - \tau_{1,0}$ and

$$\tau_{2,t} = \sum_{g=1}^{9}\left(\tau_{1,t-1}B_{g,t-1} + \tau_{2,t-1}K_{g,t-1}\right)S_g \text{ with } t > 0.$$

New (female) adults in period $t$ as a proportion of the total of population in period $t$,

$$\tau_{3,t} = \frac{\tau_{1,t}}{\tau_{1,t} + \tau_{2,t}}.$$

Adult females surviving from the preceding period to the start of period $t$ as a proportion of the population in period $t$, $\tau_{4,t} = \dfrac{\tau_{2,t}}{\tau_{1,t} + \tau_{2,t}}.$

Population of adult females in period $t$ as a proportion of the baseline population, $\tau_{5,t} = \tau_{1,t} + \tau_{2,t}$.

The proportion of new adults in each cycle that have genotype $g$ in the baseline population, $B_{g,0}$ is calculated based on specified initial allele proportions, assuming Hardy-Weinberg equilibrium (independently for each locus).

The proportion of new adults in time period $t$ (with $t > 0$) that have genotype $g$ is calculated as the proportion of eggs with that genotype from $\varpi$ periods ago, where $\varpi$ represents the number of model time periods required for development from egg to new adults. For early time periods, before $t > \varpi$, the baseline egg genotype proportions are used ($t = 0$), giving $B_{g,t} = E_{g,t-\min(t,\varpi)}$.

The proportion of surviving adult females at start of time period $t$ that have genotype $g$, $K_{g,t}$, for the baseline population is the same as the proportion of eggs with that genotype, giving $K_{g,0} = B_{g,0}$, since in the absence of an intervention there is assumed to be no difference in survival or fecundity between the phenotypes associated with each genotype. Note that for simplicity, this assumption is applied to the baseline population even when a cost of resistance is considered once an intervention is introduced. Following introduction of an intervention, the proportion of surviving adult females at start of time period $t$ that have genotype $g$ is calculated as the proportion of females with genotype $g$ that survive from the previous period, divided by the total proportion surviving, with all genotypes, giving

$$K_{g,t} = \frac{W_{g,t-1}S_g}{\psi_t} \qquad t > 0.$$

$W_{g,t}$, the proportion of females with genotype $g$ at start of time period $t$ is calculated as the sum of new adults and surviving adults with that genotype at that time, $W_{g,t} = B_{g,t}\tau_{3,t} + K_{g,t}\tau_{4,t}$.

$\psi_t$, the proportion of the population alive at start of time period $t - 1$ that survive to start of time period $t$ is calculated as the sum of the proportions with each genotype alive at start of time period $t - 1$ multiplied by the one-period survival probability associated with that genotype $\psi_t = \sum_{i=1}^{9}\left(W_{i,t-1}S_i\right)$.

The model assumes that females mate once, as new adults, with newly emerged adult males. The normalized number of eggs with genotype $g$ laid in period $t$, $\theta_{g,t}$, is therefore calculated

as the total for all previous periods of the proportion of eggs of genotype $g$ resulting from mating with new males in that period, for females of each genotype, multiplied by the probability for females of that genotype of surviving to period $t$:

$$t \quad \theta_{g,t} = \sum_{i=1}^{t} \left( \sum_{j=1}^{9} (B_{j,i} Z_{g,j,i} A_{j,i} S_j^{t-i}) \tau_{1,i} \right).$$

The proportion of the eggs laid in time period $t$ that have genotype $g$ is calculated as

$$E_{g,t} = \frac{\theta_{g,t}}{\delta_t}.$$

The total (normalized) number of eggs laid in period $t$ is calculated as $\delta_t = \sum_{g=1}^{9} \theta_{g,t}$.

Some combinations of maternal and offspring genotypes are not possible, so the proportion of eggs of genotype $g$ produced by females with genotype $j$ that are new adults in time period $t$, $Z_{g,j,t}$, is zero for all time periods for some genotypes.

$$
\begin{aligned}
Z_{1,i,t} &= 0 & \{i = 3,6,7,8,9\} \\
Z_{2,i,t} &= 0 & \{i = 7,8,9\} \\
Z_{3,i,t} &= 0 & \{i = 1,7,8,9\} \\
Z_{4,i,t} &= 0 & \{i = 3,6,9\} \\
Z_{6,i,t} &= 0 & \{i = 1,4,7\} \\
Z_{7,i,t} &= 0 & \{i = 1,2,3,7,9\} \\
Z_{8,i,t} &= 0 & \{i = 1,2,3\} \\
Z_{9,i,t} &= 0 & \{i = 1,2,3,4,7\}
\end{aligned}
$$

For other combinations values are calculated as follows:

| | |
|---|---|
| $Z_{1,1,t} = D_{1,t}$ | % eggs which are genotype rrdd in time period $t$ for new females with genotype rrdd |
| $Z_{1,2,t} = 0.5 D_{1,t}$ | % eggs which are genotype rrdd in time period $t$ for new females with genotype rRdd |
| $Z_{1,4,t} = 0.5 D_{1,t}$ | % eggs which are genotype rrdd in time period $t$ for new females with genotype rrdD |
| $Z_{1,5,t} = 0.25 D_{1,t}$ | % eggs which are genotype rrdd in time period $t$ for new females with genotype rRdD |
| $Z_{2,1,t} = D_{3,t}$ | % eggs which are genotype rRdd in time period $t$ for new females with genotype rrdd |
| $Z_{2,2,t} = 0.5 (D_{1,t} + D_{3,t})$ | % eggs which are genotype rRdd in time period $t$ for new females with genotype rRdd |
| $Z_{2,3,t} = D_{1,t}$ | % eggs which are genotype rRdd in time period $t$ for new females with genotype RRdd |
| $Z_{2,4,t} = 0.5 D_{3,t}$ | % eggs which are genotype rRdd in time period $t$ for new females with genotype rrdD |
| $Z_{2,5,t} = 0.25 (D_{1,t} + D_{3,t})$ | % eggs which are genotype rRdd in time period $t$ for new females with genotype rRdD |
| $Z_{2,6,t} = 0.5 D_{1,t}$ | % eggs which are genotype rRdd in time period $t$ for new females with genotype RRdD |
| $Z_{3,2,t} = 0.5 D_{3,t}$ | % eggs which are genotype RRdd in time period $t$ for new females with genotype rRdd |
| $Z_{3,3,t} = D_{3,t}$ | % eggs which are genotype RRdd in time period $t$ for new females with genotype RRdd |
| $Z_{3,5,t} = 0.25 D_{3,t}$ | % eggs which are genotype RRdd in time period $t$ for new females with genotype rRdD |
| $Z_{3,6,t} = 0.5 D_{3,t}$ | % eggs which are genotype RRdd in time period $t$ for new females with genotype RRdD |

*continued on next page*

| | |
|---|---|
| $Z_{4,1,t} = D_{2,t}$ | % eggs which are genotype rrdD in time period $t$ for new females with genotype rrdd |
| $Z_{4,2,t} = 0.5D_{2,t}$ | % eggs which are genotype rrdD in time period $t$ for new females with genotype rRdd |
| $Z_{4,4,t} = 0.5(D_{1,t} + D_{2,t})$ | % eggs which are genotype rrdD in time period $t$ for new females with genotype rrdD |
| $Z_{4,5,t} = 0.25(D_{1,t} + D_{2,t})$ | % eggs which are genotype rrdD in time period $t$ for new females with genotype rRdD |
| $Z_{4,7,t} = D_{1,t}$ | % eggs which are genotype rrdD in time period $t$ for new females with genotype rrDD |
| $Z_{4,8,t} = 0.5D_{1,t}$ | % eggs which are genotype rrdD in time period $t$ for new females with genotype rRDD |
| $Z_{5,1,t} = D_{4,t}$ | % eggs which are genotype rRdD in time period $t$ for new females with genotype rrdd |
| $Z_{5,2,t} = 0.5(D_{2,t} + D_{4,t})$ | % eggs which are genotype rRdD in time period $t$ for new females with genotype rRdd |
| $Z_{5,3,t} = D_{2,t}$ | % eggs which are genotype rRdD in time period $t$ for new females with genotype RRdd |
| $Z_{5,4,t} = 0.5(D_{3,t} + D_{4,t})$ | % eggs which are genotype rRdD in time period $t$ for new females with genotype rrdD |
| $Z_{5,5,t} = 0.25(D_{1,t} + D_{2,t} + D_{3,t} + D_{4,t})$ | % eggs which are genotype rRdD in time period $t$ for new females with genotype rRdD |
| $Z_{5,6,t} = 0.5(D_{1,t} + D_{2,t})$ | % eggs which are genotype rRdD in time period $t$ for new females with genotype RRdD |
| $Z_{5,7,t} = D_{3,t}$ | % eggs which are genotype rRdD in time period $t$ for new females with genotype rrDD |
| $Z_{5,8,t} = 0.5(D_{1,t} + D_{3,t})$ | % eggs which are genotype rRdD in time period $t$ for new females with genotype rRDD |
| $Z_{5,9,t} = D_{1,t}$ | % eggs which are genotype rRdD in time period $t$ for new females with genotype RRDD |
| $Z_{6,2,t} = 0.5D_{4,t}$ | % eggs which are genotype RRdD in time period $t$ for new females with genotype rRdd |
| $Z_{6,3,t} = D_{4,t}$ | % eggs which are genotype RRdD in time period $t$ for new females with genotype RRdd |
| $Z_{6,5,t} = 0.25(D_{3,t} + D_{4,t})$ | % eggs which are genotype RRdD in time period $t$ for new females with genotype rRdD |
| $Z_{6,6,t} = 0.5(D_{3,t} + D_{4,t})$ | % eggs which are genotype RRdD in time period $t$ for new females with genotype RRdD |
| $Z_{6,8,t} = 0.5D_{3,t}$ | % eggs which are genotype RRdD in time period $t$ for new females with genotype rRDD |
| $Z_{6,9,t} = D_{3,t}$ | % eggs which are genotype RRdD in time period $t$ for new females with genotype RRDD |
| $Z_{7,4,t} = 0.5D_{2,t}$ | % eggs which are genotype rrDD in time period $t$ for new females with genotype rrdD |
| $Z_{7,5,t} = 0.25D_{2,t}$ | % eggs which are genotype rrDD in time period $t$ for new females with genotype rRdD |
| $Z_{7,7,t} = D_{2,t}$ | % eggs which are genotype rrDD in time period $t$ for new females with genotype rrDD |
| $Z_{7,8,t} = 0.5D_{2,t}$ | % eggs which are genotype rrDD in time period $t$ for new females with genotype rRDD |
| $Z_{8,4,t} = 0.5D_{4,t}$ | % eggs which are genotype rRDD in time period $t$ for new females with genotype rrDD |
| $Z_{8,5,t} = 0.25(D_{2,t} + D_{4,t})$ | % eggs which are genotype rRDD in time period $t$ for new females with genotype rRdD |

| | |
|---|---|
| $Z_{8,6,t} = 0.5D_{2,t}$ | % eggs which are genotype rRDD in time period $t$ for new females with genotype RRdD |
| $Z_{8,7,t} = D_{4,t}$ | % eggs which are genotype rRDD in time period $t$ for new females with genotype rrDD |
| $Z_{8,8,t} = 0.5(D_{2,t} + D_{4,t})$ | % eggs which are genotype rRDD in time period $t$ for new females with genotype rRDD |
| $Z_{8,9,t} = D_{2,t}$ | % eggs which are genotype rRDD in time period $t$ for new females with genotype RRDD |
| $Z_{9,5,t} = 0.25D_{4,t}$ | % eggs which are genotype RRDD in time period $t$ for new females with genotype rRdD |
| $Z_{9,6,t} = 0.5D_{4,t}$ | % eggs which are genotype RRDD in time period $t$ for new females with genotype RRdD |
| $Z_{9,8,t} = 0.5D_{4,t}$ | % eggs which are genotype RRDD in time period $t$ for new females with genotype rRDD |
| $Z_{9,9,t} = D_{4,t}$ | % eggs which are genotype RRDD in time period $t$ for new females with genotype RRDD |

The average number of eggs laid during time period $t$ by females with genotype $g$ that are alive at the start of time period $t$, $A_{g,t}$, is calculated as the normalized number of eggs per mosquito per lay, multiplied by the probability that females of genotype $g$ will survive one time period. For the baseline population, this gives $A_{g,0} = \beta_0 L$, and for the population post introduction of an intervention, $A_{g,t} = S_g L \quad t>0$ .

The model is interested in comparative and proportionate rather than absolute values for egg production, for convenience therefore, we use a normalized value for the number of eggs assumed per mosquito per successful lay, represented by $L$.

$X_t$ represents the newly mated females in time period $t$ as % baseline population. For the baseline population, the genotype proportions in females and their eggs is assumed to be constant over time. Females surviving from the baseline population will therefore contribute eggs to subsequent periods with the same genotype proportions in the eggs for each maternal genotype, irrespective of age. For this purpose, therefore, as all females will have the same characteristics as newly mated females, we have $X_0 = 1$. Thereafter $X_t = \tau_{1,t}$, with $t>0$.

Mating males in time period $t$ are assumed to have the genotype proportions of newly emerged adults in period $t$. With allele pairs indexed as 1=rd, 2=rD, 3=Rd, 4=RD, the proportion of male gametes with allele pair $w$ in time period $t$, $D_{w,t}$, is therefore calculated as:

$$
\begin{aligned}
D_{1,t} &= B_{1,t} + 0.5(B_{2,t} + B_{4,t}) + 0.25B_{5,t} \\
D_{2,t} &= B_{7,t} + 0.5(B_{4,t} + B_{8,t}) + 0.25B_{5,t} \\
D_{3,t} &= B_{3,t} + 0.5(B_{2,t} + B_{6,t}) + 0.25B_{5,t} \\
D_{4,t} &= B_{9,t} + 0.5(B_{8,t} + B_{6,t}) + 0.25B_{5,t}
\end{aligned}
$$

The proportion of alleles at the resistance locus in the population at time period $t$ that are resistance alleles, $G_{1,t}$, is calculated as $G_{1,t} = W_{3,t} + W_{6,t} + W_{9,t} + 0.5(W_{2,t} + W_{5,t} + W_{8,t})$.

The proportion of alleles at the deflection locus in the population at time period $t$ that are deflection alleles, $G_{2,t}$, is calculated as $G_{2,t} = 0.5(W_{4,t} + W_{5,t} + W_{6,t}) + W_{7,t} + W_{8,t} + W_{9,t}$.

The proportion of the adult female population in time period $t$ that have resistant (non-deflected) phenotypes, $P_{1,t}$, is the sum of those with genotypes rRdd or RRdd, $P_{1,t} = W_{2,t} + W_{3,t}$.

The proportion of the adult female population in time period $t$ that have deflected phenotypes, $P_{2,t}$, is the sum of those with genotypes having Dd or DD in combination with any allele combination at the resistance locus, $P_{2,t} = \sum_{i=4}^{9} W_{i,t}$.

The proportion of the adult female population in time period $t$ that have susceptible, non-deflected phenotypes, $P_{3,t}$, is equal to the proportion with rrdd genotypes: $P_{3,t} = 1 - P_{1,t} - P_{2,t}$.

The proportion of the adult female population in time period $t$ that have two susceptible alleles (rr) at the resistance locus, $V_{1,t}$, is calculated as $V_{1,t} = W_{1,t} + W_{4,t} + W_{7,t}$.

The proportion of the adult female population in time period $t$ that have at least one resistance allele (rR, RR) at the resistance locus, $V_{2,t}$, is calculated as $V_{2,t} = W_{2,t} + W_{3,t} + W_{5,t} + W_{6,t} + W_{8,t} + W_{9,t}$.

The proportion of the adult female population in time period $t$ that have two non-deflection alleles (dd) at the deflection locus, $V_{3,t}$, is calculated as $V_{3,t} = W_{1,t} + W_{2,t} + W_{3,t}$.

The proportion of the adult female population in time period $t$ that have at least one deflection allele (rD, DD) at the resistance locus, $V_{4,t}$, is calculated as $V_{4,t} = \sum_{i=4}^{9} W_{i,t}$.

## Appendix 2

### Derivation of *Equations 1 and 2* and *Expressions 1 and 2*

Because the fitness of the individuals carrying deflection and non-deflection alleles depends on whether they are also carrying resistance alleles, and because resistance alleles may originate from the same parent as the relevant deflection allele or from the paired zygote, we consider the average fitness of the phenotypes into which deflection and non-deflection alleles will be inherited, based on the proportions of each possible allele pair in the population of zygotes, assuming the same proportions in gametes of mating males and newly emerged females.

Consider the allele pairs in gametes in the mating adult population:

$[dr]$ = proportion of deflection locus alleles which are non-deflection alleles paired with non-resistance alleles in gametes

$[dR]$ = proportion of deflection locus alleles which are non-deflection alleles paired with resistance alleles in gametes

$[Dr]$ = proportion of deflection locus alleles which are deflection alleles paired with non-resistance alleles in gametes

$[DR]$ = proportion of deflection locus alleles which are deflection alleles paired with resistance alleles in gametes

$[d-]$ = proportion of deflection locus alleles in gametes which are non-deflection alleles

$[D-]$ = proportion of deflection locus alleles in gametes which are deflection alleles

$[-r]$ = proportion of deflection locus alleles which are paired with susceptible alleles in gametes

$[-R]$ = proportion of deflection locus alleles which are paired with resistance alleles in gametes

$$[d-]+[D-]=[-r]+[-R]=1$$

Let $[S]_d$ represent the proportion of non-deflection alleles inherited into offspring with genotypes corresponding to susceptible, non-deflected phenotypes:

$$[S]_d=\frac{[dr]^2}{[d-]}.$$

Let $[R]_d$ represent the proportion of non-deflection alleles inherited into offspring with genotypes corresponding to resistant, non-deflected phenotypes:

$$[R]_d=\frac{[dr][dR]+[dR][dr]+[dR][dR]}{[d-]}=\frac{[d-][dR]+[dR][dr]}{[d-]}=[dR]+\frac{[dR][dr]}{[d-]}.$$

Let $[DR]_d$ represent the proportion of non-deflection alleles inherited into offspring with genotypes corresponding to resistant, deflected phenotypes:

$$[DR]_d=\frac{[dr][DR]+[dR][Dr]+[dR][DR]}{[d-]}=\frac{[d-][DR]+[dR][Dr]}{[d-]}=[DR]+\frac{[dR][Dr]}{[d-]}.$$

Let $[D]_d$ represent the proportion of non-deflection alleles inherited into offspring with genotypes corresponding to non-resistant, deflected phenotypes:

$$[D]_d=\frac{[dr][Dr]}{[d-]}.$$

Let $F_S$ represent the fitness of susceptible, non-deflected phenotypes.

Let $F_R$ represent the fitness of resistant, non-deflected phenotypes.

Let $F_{RD}$ represent the fitness of resistant, deflected phenotypes.

Let $F_D$ represent the fitness of non-resistant, deflected phenotypes.

Let $\overline{F}_D$ represent the average fitness of offspring into which deflection alleles are inherited.

Let $[DR]_D$ represent the proportion of deflection alleles inherited into offspring with genotypes corresponding to resistant, deflected phenotypes:

$$[DR]_D = \frac{[Dr][dR] + [Dr][DR] + [DR][DR] + [DR][Dr] + [DR][dR] + [DR][dr]}{[D-]}$$

$$= [-R] + \frac{[DR][-r]}{[D-]}$$

Let $[D]_D$ represent the proportion of deflection alleles inherited into offspring with genotypes corresponding to non-resistant, deflected phenotypes:

$$[D]_D = \frac{[Dr][dr] + [Dr][Dr]}{[D-]}$$

$$= \frac{[Dr][-r]}{[D-]}$$

$$\overline{F}_D = [DR]_D F_{RD} + [D]_D F_D$$

$$= \left([-R] + \frac{[DR][-r]}{[D-]}\right) F_{RD} + \frac{[Dr][-r]}{[D-]} F_D$$

$$= F_S + \frac{[Dr][-R] + [DR]}{[D-]}(F_{RD} - F_S) + \frac{[Dr][-r]}{[D-]}(F_D - F_S)$$

This is Equation 1.

Let $\overline{F}_d$ represent the average fitness of offspring into which non-deflection alleles are inherited:

$$\overline{F}_d = [S]_d F_S + [R]_d F_R + [RD]_d F_{RD} + [D]_d F_D$$

$$\overline{F}_d = \frac{[dr]^2}{[d-]} F_S + \left([dR] + \frac{[dR][dr]}{[d-]}\right) F_R + \left([DR] + \frac{[dR][Dr]}{[d-]}\right) F_{RD} + \frac{[dr][Dr]}{[d-]} F_D$$

$$\overline{F}_d = F_S + \left([dR] + \frac{[dR][dr]}{[d-]}\right)(F_R - F_S) + \frac{[dr][Dr]}{[d-]}(F_D - F_S) + \left([DR] + \frac{[dR][Dr]}{[d-]}\right)(F_{RD} - F_S)$$

This is Equation 2.

Deflection alleles will spread when the average fitness of offspring into which deflection alleles are inherited is higher than that of offspring into which non-deflection alleles are inherited:

$$\overline{F}_D > \overline{F}_d \leftrightarrow F_S + \frac{[Dr][-R] + [DR]}{[D-]}(F_{RD} - F_S) + \frac{[Dr][-r]}{[D-]}(F_D - F_S)$$

$$> F_S + \left([dR] + \frac{[dR][dr]}{[d-]}\right)(F_R - F_S) + \left([DR] + \frac{[dR][Dr]}{[d-]}\right)(F_{RD} - F_S) + \frac{[dr][Dr]}{[d-]}(F_D - F_S)$$

This can be rearranged as follows to give Expression 1.

Lynch and Boots. eLife 2016;5:e15416. DOI: 10.7554/eLife.15416

$$\overline{F}_D > \overline{F}_d \leftrightarrow \frac{[Dr][-R]+[DR]}{[D-]}(F_{RD}-F_S)+\frac{[Dr][-r]}{[D-]}(F_D-F_S)$$

$$> \left([dR]+\frac{[dR][dr]}{[d-]}\right)(F_R-F_S)+\left([DR]+\frac{[dR][Dr]}{[d-]}\right)(F_{RD}-F_S)+\frac{[dr][Dr]}{[d-]}(F_D-F_S)$$

$$\overline{F}_D > \overline{F}_d \leftrightarrow \frac{[Dr][-R]+[DR]}{[D-]}(F_{RD}-F_D)+\frac{[Dr][-r]+[Dr][-R]+[DR]}{[D-]}(F_D-F_S)$$

$$> \left([dR]+\frac{[dR][dr]}{[d-]}\right)(F_R-F_S)+\left([DR]+\frac{[dR][Dr]}{[d-]}\right)(F_{RD}-F_D)$$

$$+\left(\frac{[dr][Dr]}{[d-]}+[DR]+\frac{[dR][Dr]}{[d-]}\right)(F_D-F_S)$$

$$\overline{F}_D > \overline{F}_d \leftrightarrow \frac{[Dr][-R]+[DR]}{[D-]}(F_{RD}-F_D)+(F_D-F_S)$$

$$> \left([dR]+\frac{[dR][dr]}{[d-]}\right)(F_R-F_S)+[D-](F_D-F_S)+\left([DR]+\frac{[Dr][dR]}{[d-]}\right)(F_{RD}-F_D)$$

$$\overline{F}_D > \overline{F}_d \leftrightarrow (F_D-F_S)$$

$$> \left([dR]+\frac{[dR][dr]}{[d-]}\right)(F_R-F_S)+[D-](F_D-F_S)+\left([DR]+\frac{[Dr][dR]}{[d-]}-\frac{[Dr][-R]+[DR]}{[D-]}\right)(F_{RD}-F_D)$$

$$\overline{F}_D > \overline{F}_d \leftrightarrow (F_D-F_S) >$$

$$[D-](F_D-F_S)+[dR]\left(1+\frac{[dr]}{[d-]}\right)(F_R-F_S)$$

$$+\left([DR]\left(1-\frac{1}{[D-]}\right)+[Dr]\left(\frac{[dR]}{[d-]}-\frac{[-R]}{[D-]}\right)\right)(F_{RD}-F_D)$$

$$\overline{F}_D > \overline{F}_d \leftrightarrow (F_D-F_S) > \left([D-]+[dR]+\frac{[dR][dr]}{[d-]}\right)(F_D-F_S)+[dR]\left(1+\frac{[dr]}{[d-]}\right)(F_R-F_D)$$

$$+\left([DR]\left(1-\frac{1}{[D-]}\right)+[Dr]\left(\frac{[dR]}{[d-]}-\frac{[-R]}{[D-]}\right)\right)(F_{RD}-F_D)$$

$$[dR]+\frac{[dR][dr]}{[d-]}=[dR]+\frac{[dR][dr]}{[dR]+[dr]}=\frac{[dR]([dR]+[dr])+[dR][dr]}{[dR]+[dr]}$$

$$=\frac{[dR]^2+[dR][dr]+[dR][dr]}{[dR]+[dr]}=\frac{([dR]+[dr])^2-[dr]^2}{[dR]+[dr]}=[d-]-\frac{[dr]^2}{[d-]}$$

$$\therefore \overline{F}_D > \overline{F}_d \leftrightarrow (F_D-F_S) > \left([D-]+[d-]-\frac{[dr]^2}{[d-]}\right)(F_D-F_S)+[dR]\left(1+\frac{[dr]}{[d-]}\right)(F_R-F_D)$$

$$+\left([DR]\left(1-\frac{1}{[D-]}\right)+[Dr]\left(\frac{[dR]}{[d-]}-\frac{[-R]}{[D-]}\right)\right)(F_{RD}-F_D)$$

$$\overline{F}_D > \overline{F}_d \leftrightarrow (F_D-F_S) > \left(1-\frac{[dr]^2}{[d-]}\right)(F_D-F_S)+[dR]\left(1+\frac{[dr]}{[d-]}\right)(F_R-F_D)$$

$$+\left([DR]\left(1-\frac{1}{[D-]}\right)+[Dr]\left(\frac{[dR]}{[d-]}-\frac{[-R]}{[D-]}\right)\right)(F_{RD}-F_D)$$

$$\overline{F}_D > \overline{F}_d \leftrightarrow \frac{[dr]^2}{[d-]}(F_D-F_S) > [dR]\left(1+\frac{[dr]}{[d-]}\right)(F_R-F_D)$$

$$+\left([DR]\left(1-\frac{1}{[D-]}\right)+[Dr]\left(\frac{[dR]}{[d-]}-\frac{[-R]}{[D-]}\right)\right)(F_{RD}-F_D)$$

Which is Expression 1

The notation and assumptions used to develop Expression 2, the inequality indicating when resistance alleles will be inherited into offspring with an average fitness higher than susceptible alleles, are similar to those used for the equivalent expression with respect to deflection alleles.

Consider the allele pairs in gametes in the mating adult population:

$[Rd]$ = proportion of resistance locus alleles which are resistance alleles paired with non-deflection alleles in gametes

$[RD]$ = proportion of resistance locus alleles which are resistance alleles paired with deflection alleles in gametes

$[rd]$ = proportion of resistance locus alleles which are susceptible alleles paired with non-deflection alleles in gametes

$[rD]$ = proportion of resistance locus alleles which are susceptible alleles paired with deflection alleles in gametes

$[R-]$ = proportion of resistance locus alleles in gametes which are resistance alleles

$[r-]$ = proportion of resistance locus alleles in gametes which are susceptible alleles

$[-d]$ = proportion of resistance locus alleles which are paired with non-deflection alleles in gametes

$[-D]$ = proportion of resistance locus alleles which are paired with deflection alleles in gametes

$$[r-]+[R-]=[-d]+[-D]=1$$

Let $\overline{F}_R$ represent the average fitness of offspring into which resistance alleles are inherited:

$$\overline{F}_R = \frac{1}{[R-]}\left(\begin{array}{l}[Rd][Rd]F_R+[Rd][RD]F_{RD}+[Rd][rd]F_R+[Rd][rD]F_{RD}\\+[RD][Rd]F_{RD}+[RD][RD]F_{RD}+[RD][rd]F_{RD}+[RD][rD]F_{RD}\end{array}\right)$$

$$\overline{F}_R = \frac{1}{[R-]}\left(\begin{array}{l}([Rd][Rd]+[Rd][rd])F_R\\+([Rd][RD]+[Rd][rD]+[RD][Rd]+[RD][RD]+[RD][rd]+[RD][rD])F_{RD}\end{array}\right)$$

$$\overline{F}_R = \frac{1}{[R-]}([Rd][-d]F_R+([Rd][-D]+[RD])F_{RD})$$

$$\overline{F}_R = \frac{1}{[R-]}([Rd][-d]F_R+([R-][-D]+[RD][-d])F_{RD})$$

$$\overline{F}_R = \frac{[Rd][-d]}{[R-]}(F_R-F_S)+\left([-D]+\frac{[RD][-d]}{[R-]}\right)(F_{RD}-F_S)+F_S$$

The average fitness of offspring into which susceptible alleles are inherited, $\overline{F}_r$, is:

$$\overline{F}_r = \frac{1}{[r-]}\left(\begin{array}{l}[rd][Rd]F_R+[rd][RD]F_{RD}+[rd][rd]F_S+[rd][rD]F_D\\+[rD][Rd]F_{RD}+[rD][RD]F_{RD}+[rD][rd]F_D+[rD][rD]F_D\end{array}\right)$$

$$\overline{F}_r = \frac{1}{[r-]}\left(\begin{array}{l}[rd][Rd]F_R+([rd][RD]+[rD][Rd]+[rD][RD])F_{RD}\\+[rd][rd]F_S+([rd][rD]+[rD][rd]+[rD][rD])F_D\end{array}\right)$$

$$\overline{F}_r = \frac{[rd][Rd]}{[r-]}F_R+\left([RD]+\frac{[rD][Rd]}{[r-]}\right)F_{RD}+\frac{[rd][rd]}{[r-]}F_S+\left([rD]+\frac{([rd][rD])}{[r-]}\right)F_D$$

$$\overline{F}_r = \frac{[rd][Rd]}{[r-]}(F_R-F_S)+\left([RD]+\frac{[rD][Rd]}{[r-]}\right)(F_{RD}-F_S)+\left([rD]+\frac{([rd][rD])}{[r-]}\right)(F_D-F_S)+F_S$$

In order for the proportion of resistance alleles in the population to increase, we need the average fitness of the offspring into which resistance alleles are inherited to be greater than the average fitness of the offspring into which susceptible alleles are inherited. This is true when the following inequality applies:

$$\overline{F}_R > \bar{F}_r \leftrightarrow \frac{[Rd][-d]}{[R-]}(F_R - F_S) + \left([-D] + \frac{[RD][-d]}{[R-]}\right)(F_{RD} - F_S) + F_S$$

$$- \frac{[rd][Rd]}{[r-]}(F_R - F_S) - \left([RD] + \frac{[rD][Rd]}{[r-]}\right)(F_{RD} - F_S) - \left([rD] + \frac{([rd][rD])}{[r-]}\right)(F_D - F_S) - F_S > 0$$

$$\leftrightarrow \left(\frac{[Rd][-d]}{[R-]} - \frac{[Rd][rd]}{[r-]}\right)(F_R - F_D) + \left(\frac{[Rd][-d]}{[R-]} - \frac{[Rd][rd]}{[r-]} - [rD] - \frac{([rd][rD])}{[r-]}\right)(F_D - F_S)$$

$$+ \left([rD] + \frac{[RD][-d]}{[R-]} - \frac{[rD][Rd]}{[r-]}\right)(F_{RD} - F_R) + \left([rD] + \frac{[RD][-d]}{[R-]} - \frac{[rD][Rd]}{[r-]}\right)(F_R - F_D)$$

$$+ \left([rD] + \frac{[RD][-d]}{[R-]} - \frac{[rD][Rd]}{[r-]}\right)(F_D - F_S) > 0$$

$$\leftrightarrow \left(\frac{[Rd][-d]}{[R-]} - \frac{[Rd][rd]}{[r-]} + [rD] + \frac{[RD][-d]}{[R-]} - \frac{[rD][Rd]}{[r-]}\right)(F_R - F_D)$$

$$+ \left(\frac{[Rd][-d]}{[R-]} - \frac{[Rd][rd]}{[r-]} - [rD] - \frac{([rd][rD])}{[r-]} + [rD] + \frac{[RD][-d]}{[R-]} - \frac{[rD][Rd]}{[r-]}\right)(F_D - F_S)$$

$$+ \left([rD] + \frac{[RD][-d]}{[R-]} - \frac{[rD][Rd]}{[r-]}\right)(F_{RD} - F_R) > 0$$

$$\leftrightarrow ([r-])(F_R - F_D) + \left([rd] - \frac{([rd][rD])}{[r-]}\right)(F_D - F_S)$$

$$+ \left([rD] + \frac{[RD][-d]}{[R-]} - \frac{[rD][Rd]}{[r-]}\right)(F_{RD} - F_R) > 0$$

$$\leftrightarrow [rd](F_R - F_S) + [rD](F_R - F_D)$$

$$+ \left([rD] + \frac{[RD][-d]}{[R-]} - \frac{[rD][Rd]}{[r-]}\right)(F_{RD} - F_R) > \frac{([rd][rD])}{[r-]}(F_D - F_S)$$

This is Expression 2.

