## [Decision Letter]

Thank you for submitting your article "Evolution leading to sustainable malaria control through evolved spatial repellents" for consideration by *eLife*. Your article has been favorably evaluated by Diethard Tautz as the Senior Editor and three reviewers: Fred Gould (Reviewer #2), Gerry Kileen (Reviewer #3), and Ben Cooper (Reviewer #1), who is a member of our Board of Reviewing Editors.

The reviewers have discussed the reviews with one another and the Reviewing Editor has drafted this decision to help you prepare a revised submission.

Summary:

This paper proposes a novel strategy for sustained malaria control: starting with a partially effective repellent (effective only against a fraction of the local mosquito population) and using evolution to generate a highly effective spatial repellent. The idea is to combine use of this repellent (referred to as the ESR – the evolved spatial repellent) to stop some proportion of mosquitoes entering human dwellings where a highly effective insecticide (ideally with low contact repellence) is used inside. Those mosquitoes which are repelled, it is argued, will under plausible conditions have selective advantages (because they won't encounter the insecticide) and genes encoding the repellence behaviour will be likely to spread through the mosquito population, leading to an increasingly effective spatial repellent.

All three reviewers found this work interesting and potentially important and felt that, if appropriate revisions are made, it should be published. Reviewer 3 commented that "this piece brings an invaluable idea to the table at a great time, exactly when several new technologies with the right properties to act on these principles are becoming available" but felt more needs to be done to ensure that this work reaches out to the field-based community in language they can relate to more easily.

Essential revisions:

1) The work needs to more fully acknowledge related work and to place in historical context. In particular, it was suggested that the authors look over other papers that have built on the Gould 1984 paper by providing more general analysis and by adapting it to specific cases in agriculture.

Some relevant references are:

Kennedy, G. G., F. Gould, O. M. B. de Ponti, and R. E. Stinner. 1987. Ecological, agricultural, genetic, and commercial consideration in the deployment of insect-resistant germ plasm. Environ. Entomol. 16: 327-338.

Castillo-Chavez, C., S. A. Levin, and F. Gould. 1988. Physiological and behavioral adaptation to varying environments: a mathematical model. Evolution 42: 986-994.

Jongsma, M. A., F. Gould, M. Legros, Y. M. Yang, J. J. A. van Loon, M. Dicke. 2010. Insect oviposition behavior affects the evolution of adaptation to Bt crops: consequences for refuge policies. Evol. Ecol. 24:1017-1030.

Gatton, M.L., N. Chitnis, T. Churcher, M. J. Donnelly, A. C. Ghani, H. C. J. Godfray, F. Gould, I. Hastings, J. Marshall, H. Ranson, M. Rowland, J. Shaman, S. W. Lindsay. 2013. The Importance of Mosquito Behavioural Adaptations to Malaria Control in Africa. Evolution. 67:1218-1230.

It was also noted that this work extends the arguments made by others in relation to combinations of irritants and lethal insecticides (PLoS One 9: e95640), to include vapor-phase repellents applied inside houses. This preceding work should be overtly and fairly acknowledged in the background and Discussion sections.

2) The manuscript should be revised to increase accessibility and broad appeal, particularly to the field-based community. Changes suggested include:

Revising the title to something that is more likely to reach out to a wider audience including non-theoreticians. In particular, the term "evolved repellents" appears to be a term that many potential readers may not be familiar with.

A punchier Conclusions section and a slightly more direct Abstract in language accessible to non-theoreticians would be of use.

Consistent use of common specific wording used widely by epidemiologists and entomologists alike. For example, instead of "infectious bite values" use "entomological inoculation rate (EIR)" (assuming that is what is meant) or the "rate of human exposure to infectious bites".

3) In the second paragraph of the Modelling section: There have been numerous studies demonstrating shifting of mosquitoes to feeding outdoors by indoor protective measures like bed nets, and the reduced sporozoite rates observed (See Lengeler et al. Cochrane review as just one example) cannot be unambiguously attributed to reduced acquisition of infections, and are more likely to arise from reduced survival or feeding frequency (PLoS Med 4: e229, Malaria Journal 20: 207). There is no biological reason to believe individuals would be any less infectious outdoors than indoors, so this is an implausible set of scenarios to explore that should be removed in the interest of brevity. It would be preferable to see the impact of deflection on other important life history parameters like survival, human blood index and feeding frequency explicitly outlined here, as they have been by other preceding models, such as Trans Roy Soc Trop Med 110: 107.

4) The third paragraph of the Introduction describes a view of malaria vector behaviour and transmission system properties based on the peculiar characteristics of a small number (albeit disproportionately important) of very human-specialized vectors from Africa in particular. See Durnez & Coosemans book chapter on residual malaria transmission, Malaria Journal 13: 330, and the 2014 guidance note on residual malaria transmission by WHO for a more inclusive view of the diversity of malaria transmission. This partial (but substantive) limitation to their relevance needs to be overtly and explicitly outlined in the Discussion, conclusions and Abstract. Also, the fact that this specific subset of transmission scenarios has been considered needs to be explained and justified in the Modelling section. This should be very easy as restricting consideration to the world's most important vectors, which enter a houses a lot, still encompasses a really a big chunk of the world's malaria and is exactly where this strategy is most likely to work.

5) Given the reliance of this piece on the assumption that sufficiently efficacious spatial repellent products exist to address these needs, the Discussion should consider emerging reports of new prototypes with improved product profiles (Parasites and Vectors 8: 322, which contrary to the third paragraph of the Introduction requires no combustion or electricity). Further encouraging updates are likely to be in the literature in the months ahead. The conclusions could apply equally to a single vapor-phase active ingredient that is only lethal if the mosquito ignores the repellent effect at sub-lethal doses for long enough to expose itself to higher lethal doses. The fluorobenzylated pyrethroids seem like an obvious example of compounds already available to which this hypothesis might well apply (US Patent 2015/0289513 A1). It was also suggested that the Discussion could expand on the broader implications by noting that an equally important implication of this hypothesis is that such indoor lethal insecticides could help preserve the efficacy of repellents to be used outdoors where contact insecticides are not feasible to use (Malaria Journal 11: 17 and 13: 146), by also using them indoors.

6) In endemic settings human infectiousness to mosquitoes spans a much narrower range than evaluated here, typically between 2 and 7% (Trans Roy Soc Hyg Trop Med 94: 472, Nature Communications 1: 108, Am J Trop Med Hyg 75 (Suppl. 2): 32 and 38). Surely this is a range exploration exercise that could be removed, to shorten a very long, sometimes repetitive paper. Use of these implausible values may have caused correspondingly implausible estimates for EIR (is this why these have been anonymized by presenting all impact estimates relative to a zero intervention scenario, rather than in absolute terms?). Please present all graphs with commonly used terms like EIR for all outputs, so that their plausibility can be transparently assessed by field specialists.

7) The authors should compare their output to other two locus models that address similar questions with specified parameters, matching the parameters and see if the results are the same. This will help to confirm that there have not been errors in implementation.

[Editors' note: further revisions were requested prior to acceptance, as described below.]

Thank you for resubmitting your work entitled "Using evolution to generate sustainable malaria control with spatial repellents" for further consideration at *eLife*. Your revised article has been favorably evaluated by Diethard Tautz as the Senior editor and a Reviewing editor.

The manuscript has been improved but there are some remaining issues that need to be addressed before acceptance, as outlined below:

1) In the second paragraph of the Modelling section it should say "see Appendix 1".

2) Appendix 2 makes it clear that *[dr], [dR]* etc. refer to proportions of alleles in gametes. Shouldn't the Modelling section also make this clear?

3) Near the end of the Modelling section it contains a couple of "it can be seen" statements that seem either incomplete, imprecise or require further justification. In the first it claims that spread of deflection will by favoured by maximising the "fitness difference between susceptible and deflected phenotypes", but this won't be true if *F_D_* < *F_S_*. Also not clear why there is no reference here to the *F_R_* – *F_D_* term. The second "it can be seen" is even less clear as there are terms on the rhs of the inequality that both increase and decrease with increasing frequency of resistance alleles. I'm prepared to believe it's generally true with plausible parameters, but I don't think it's necessarily easily seen.

4) In the fourth paragraph of the Modelling section "data become" rather than "data becomes" (also some slightly erratic punctuation throughout which no doubt will be fixed in the production process).

5) In the fifth paragraph of the Results "0.05%" – the Figure 2 caption says 0.5% – presumably the latter is correct.

---

## [Author Response]

*Essential revisions:*

1) The work needs to more fully acknowledge related work and to place in historical context. In particular, it was suggested that the authors look over other papers that have built on the Gould 1984 paper by providing more general analysis and by adapting it to specific cases in agriculture.

*Some relevant references are:*

*Kennedy, G. G., F. Gould, O. M. B. de Ponti, and R. E. Stinner. 1987. Ecological, agricultural, genetic, and commercial consideration in the deployment of insect-resistant germ plasm. Environ. Entomol. 16: 327-338.*

*Castillo-Chavez, C., S. A. Levin, and F. Gould. 1988. Physiological and behavioral adaptation to varying environments: a mathematical model. Evolution 42: 986-994.*

*Jongsma, M. A., F. Gould, M. Legros, Y. M. Yang, J. J. A. van Loon, M. Dicke. 2010. Insect oviposition behavior affects the evolution of adaptation to Bt crops: consequences for refuge policies. Evol. Ecol. 24:1017-1030.*

*Gatton, M.L., N. Chitnis, T. Churcher, M. J. Donnelly, A. C. Ghani, H. C. J. Godfray, F. Gould, I. Hastings, J. Marshall, H. Ranson, M. Rowland, J. Shaman, S. W. Lindsay. 2013. The Importance of Mosquito Behavioural Adaptations to Malaria Control in Africa. Evolution. 67:1218-1230.*

*It was also noted that this work extends the arguments made by others in relation to combinations of irritants and lethal insecticides (PLoS One 9: e95640), to include vapor-phase repellents applied inside houses. This preceding work should be overtly and fairly acknowledged in the background and Discussion sections.*

We thank the reviewers for bringing the above papers to our attention, and have included citations for them in the manuscript. We have also included citations for an illustrative range of previous modelling in relevant areas, and cited a review of mathematical modelling for malaria,[1] to provide additional context for interested readers.

*2) The manuscript should be revised to increase accessibility and broad appeal, particularly to the field-based community. Changes suggested include:*

*Revising the title to something that is more likely to reach out to a wider audience including non-theoreticians. In particular, the term "evolved repellents" appears to be a term that many potential readers may not be familiar with.*

A punchier Conclusions section and a slightly more direct Abstract in language accessible to non-theoreticians would be of use.

We have revised the title to eliminate the unfamiliar term ‘Evolved repellents’, as suggested. We have also made revisions to the Abstract and changes throughout the text in response to this and other comments which have hopefully helped to clarify and simplify some key sections of the manuscript.

*Consistent use of common specific wording used widely by epidemiologists and entomologists alike. For example, instead of "infectious bite values" use "entomological inoculation rate (EIR)" (assuming that is what is meant) or the "rate of human exposure to infectious bites".*

We take the reviewer’s point that our message needs to be in the most accessible form possible, and have made changes to the text to try to achieve this. If we make the simple assumption that human population size is not materially affected by the chosen intervention, and is stable over time, we can equate our infectious bite values to EIR and we have now explained this in the manuscript.

*3) In the second paragraph of the Modelling section: There have been numerous studies demonstrating shifting of mosquitoes to feeding outdoors by indoor protective measures like bed nets, and the reduced sporozoite rates observed (See Lengeler et al. Cochrane review as just one example) cannot be unambiguously attributed to reduced acquisition of infections, and are more likely to arise from reduced survival or feeding frequency (PLoS Med 4: e229, Malaria Journal 20: 207).*

We have amended our wording about the Ndiath result to ensure that we do not misrepresent its significance.

“The impact of deflection on malaria prevalence is determined by the proportion of mosquitoes deflected by a repellent and the probability, compared to non-deflected mosquitoes, that they will then acquire and transmit a Plasmodium infection. […] We have not explicitly assessed the additional effects of reduced transmission from outdoor-feeding infectious mosquitoes resulting from, for example, transfer to feeding on non-human hosts, but any such effects would clearly serve to further enhance the public health benefits of deflection.”

*There is no biological reason to believe individuals would be any less infectious outdoors than indoors, so this is an implausible set of scenarios to explore that should be removed in the interest of brevity. It would be preferable to see the impact of deflection on other important life history parameters like survival, human blood index and feeding frequency explicitly outlined here, as they have been by other preceding models, such as Trans Roy Soc Trop Med 110: 107.*

With apologies, this and other comments from the reviewers demonstrate that our explanation of this part of our analysis was confusing, and we have amended the manuscript to improve this. The impact of deflection on important life history parameters such as human blood index is included in the ‘transmission probability’ parameter in our model, which is the probability of transmission to the vector per feed and encompasses any difference between the probability of acquiring an infection when feeding outside rather than inside a human dwelling. As well as different likelihoods of taking a non-human feed, other possible causes of differential per feed transmission probabilities are not yet well-explored, but there is for example, interesting evidence to suggest that vector feeding times may affect the availability or transmissibility of gametocytes, and we have added some additional references regarding this. It is because of the uncertainty regarding such effects, and the potential for them to vary between species (vector and parasite) and context, that we have chosen to consider a wide range of possible transmission probabilities in our modelling.

Assumed differential vector survival with and without deflection is intrinsic to our modelling, and its effects on survival to feed, survival to reach infectiousness and survival to give each infectious bite are reflected directly through the survival parameters in the model rather than the per feed probabilities of transmission to/from the vector. We have amended the text to ensure this is clear.

*4) The third paragraph of the Introduction describes a view of malaria vector behaviour and transmission system properties based on the peculiar characteristics of a small number (albeit disproportionately important) of very human-specialized vectors from Africa in particular. See Durnez & Coosemans book chapter on residual malaria transmission, Malaria Journal 13: 330, and the 2014 guidance note on residual malaria transmission by WHO for a more inclusive view of the diversity of malaria transmission. This partial (but substantive) limitation to their relevance needs to be overtly and explicitly outlined in the Discussion, conclusions and Abstract. Also, the fact that this specific subset of transmission scenarios has been considered needs to be explained and justified in the Modelling section. This should be very easy as restricting consideration to the world's most important vectors, which enter a houses a lot, still encompasses a really a big chunk of the world's malaria and is exactly where this strategy is most likely to work.*

We have amended the text to explicitly identify that ESR is a tool with relevance in the context of public health campaigns based on indoor residual spraying (IRS), which inherently target only indoor-feeding vectors. We thank the reviewer for highlighting that this was not sufficiently clear in the manuscript.

“Malaria vectors typically feed indoors between dusk and dawn and vector control has focused on exploiting this behaviour to deliver lethal control measures against indoor-feeding mosquitoes [Yakob, Dunning and Yan, 2011], with outdoor biting viewed as unwanted behavioural resistance [Bradley et al., 2012; Russell et al., 2011; Reddy et al., 2011; Cooke et al., 2015; Mouchet, Hamon and World Health Organization, 1963].”

The reviewer identifies targeting of indoor-feeding vectors as exclusively affecting highly anthropophilic vectors, however, we would argue that other important vectors which have a high propensity to feed on non-human hosts, such as *An. arabiensis*, also show indoor feeding on human hosts [2] and are targeted by IRS. These are potentially very important targets for ESR, since the fitness costs of deflection would be expected to be lower for a vector more likely to deflect to an alternative host, making it easier to establish and maintain an ESR, whilst increasing the proportion of feeds taken on non-human hosts also reduces the probability of transmission to/from vectors, since the vectors can neither acquire nor transmit a *Plasmodium* infection during a feeding cycle utilising a non-human host.

*5) Given the reliance of this piece on the assumption that sufficiently efficacious spatial repellent products exist to address these needs, the Discussion should consider emerging reports of new prototypes with improved product profiles (Parasites and Vectors 8: 322, which contrary to the third paragraph of the Introduction requires no combustion or electricity). Further encouraging updates are likely to be in the literature in the months ahead.*

Thanks to the reviewer for bringing to our attention to this interesting and positive article by Govella et al. [3], which assesses the performance of close proximity transfluthrin-impregnated hessian strips in providing protection from mosquito bites outdoors. As discussed by Govella et al., there are still a number of issues to consider before translating this into a method for keeping mosquitoes out of properties (and see below), but we agree that this is nonetheless interesting and relevant to a consideration of currently available repellents. Transfluthrin has also been used for short periods in another study as a spatial repellent (in the sense used in our manuscript) [4]. We have therefore added references to this and some other recent articles on research into malaria vector repellents with various modes of repellent action.

“Effective action at a distance is still challenging to achieve, and commonly requires active dispersal through combustion or powered devices [Menger et al., 2016; Syafruddin et al., 2014].”

*The conclusions could apply equally to a single vapor-phase active ingredient that is only lethal if the mosquito ignores the repellent effect at sub-lethal doses for long enough to expose itself to higher lethal doses. The fluorobenzylated pyrethroids seem like an obvious example of compounds already available to which this hypothesis might well apply (US Patent 2015/0289513 A1).*

Thanks to the reviewer for this interesting comment. Review of specific compounds with potential for use as ESRs is clearly a key part of the next stage of this research, but is beyond the intended scope of the current paper.

*It was also suggested that the Discussion could expand on the broader implications by noting that an equally important implication of this hypothesis is that such indoor lethal insecticides could help preserve the efficacy of repellents to be used outdoors where contact insecticides are not feasible to use (Malaria Journal 11: 17 and 13: 146), by also using them indoors.*

We think this is a very interesting idea. The context and the specific characteristics of the chosen ESR would be critical to whether this would work. For example, naturally outdoor-feeding vectors will experience little or no selection from indoor insecticides, so ESR will have little influence on their behaviour. If a vector population is feeding both indoors and outdoors, then widespread use of an ESR outdoors would increase the relative fitness costs of deflection so might delay establishment of an ESR and/or accelerate its loss to the spread of insecticide resistance. If the overall fitness cost is low though, say a zoophagic vector with plenty of available non-human hosts, or use of outdoor repellent for personal protection by only a small number of at-risk people, then the ESR concept might indeed generate and/or maintain functionality of a compound as an outdoor repellent. This would however also be conditional on finding a compound which has the required characteristics both to protect a contained indoor space and to protect an open outdoor area and/or be used on skin or clothing. We have tried to include a suitably caveated but succinct reference to this idea in the manuscript.

*6) In endemic settings human infectiousness to mosquitoes spans a much narrower range than evaluated here, typically between 2 and 7% (Trans Roy Soc Hyg Trop Med 94: 472, Nature Communications 1: 108, Am J Trop Med Hyg 75 (Suppl. 2): 32 and 38). Surely this is a range exploration exercise that could be removed, to shorten a very long, sometimes repetitive paper. Use of these implausible values may have caused correspondingly implausible estimates for EIR (is this why these have been anonymized by presenting all impact estimates relative to a zero intervention scenario, rather than in absolute terms?).*

We particularly appreciate this comment as it has brought to our attention that we have at some point lost from the manuscript the values assumed for some base parameter values, including the assumed probability of a vector acquiring a *Plasmodium* infection when feeding indoors, which is 4%, well placed in the 2%−7% range indicated by the reviewer. With apologies for this omission, we have reinstated this and the other base parameter values in the document.

The equivalent vector infection probabilities assumed for deflected vectors feeding outdoors are 100% x 4% = 4%, 50% x 4% = 2%, 25% x 4% = 1% and 0% x 4% = 0%. Because our results are formulated in comparative rather than absolute terms, the relationship between the assumed indoor and outdoor transmission values is key. However, the way we have given this information has clearly generated confusion, so we have amended both the main text and the Figure 4 labels to make clearer the actual rather than just the relative transmission probabilities assumed.

We agree that our presentation of this issue involved some unnecessary repetition and have amended the manuscript to improve this.

As indicated in the manuscript, we formulate our results in comparative rather than absolute terms to minimize the impact of poorly quantified or location/species specific parameters on the results. The intention is to consider whether or not the proposed innovation can offer a reduction in transmission, and for this purpose comparative metrics are highly suitable, and have previously been used for evaluation of novel anti-malaria interventions and assessment of the significance of possible vector-parasite interactions [5-7]. Our assumption here is that this analysis considers the potential benefits of adding ESR to existing public health strategies, and that detailed, site-specific analysis would be carried out to establish its suitability for each given setting.

*Please present all graphs with commonly used terms like EIR for all outputs, so that their plausibility can be transparently assessed by field specialists.*

For Figure 1 the graphs show the proportion of the vector population with given genotypes and phenotypes, plotted against time. We have changed the headings slightly to ensure that this is clear. We have also corrected the *x*-axis labels for this figure, which had mysteriously acquired unwanted ‘%’ signs during the last stage of figure preparation.

For Figure 2 and Figure 3 the grid plots indicate the combinations of deflected and resistant per-cycle survival values which generate a vector population with at least 80% deflected phenotypes after 300 time periods. We have amended the figure captions to try to make the meaning of the figures as clear as possible, but we are not aware of any standard terminology that could be used here.

Figure 4 shows the number of infectious bites from the vector population per unit of time as a percentage of that without any intervention. With an additional assumption that the human population size does not change over time this metric equates to EIR expressed as a% of the EIR with no intervention. We have stated this additional assumption and changed the graph labels accordingly.

*7) The authors should compare their output to other two locus models that address similar questions with specified parameters, matching the parameters and see if the results are the same. This will help to confirm that there have not been errors in implementation.*

With the agreement of the editors we have addressed this point using a separate ‘proof of concept’ (‘POC’) model, which we created prior to building the complete model documented in Appendix 1. Although both enacted in Excel, the two models were built separately from scratch, so would not be expected to include matching typos or other corresponding errors in implementation. The POC model does not reflect the mating of females only at the start of their adult lives, and does not track *Plasmodium* infection or transmission in the vector population. We compared results from the two models for the three cases illustrated in Figure 1 and for one COR case from Figure 3. The change in genotype proportions was slower using the PM than the POC, consistent with the damping effect of reflecting the male genotypes applicable at the time of initial mating for females surviving from period to period. Otherwise the results were qualitatively consistent in all cases.

References

1) Sandip Mandal RRS, Somdatta Sinha. Mathematical models of malaria - a review. Malaria Journal. 2011;10(202).

2) Sinka ME, Bangs MJ, Manguin S, Coetzee M, Mbogo CM, Hemingway J, et al. The dominant Anopheles vectors of human malaria in Africa, Europe and the Middle East: occurrence data, distribution maps and bionomic précis. Parasites & Vectors. 2010;3(1):1-34.

3) Govella NJ, Ogoma SB, Paliga J, Chaki PP, Killeen G. Impregnating hessian strips with the volatile pyrethroid transfluthrin prevents outdoor exposure to vectors of malaria and lymphatic filariasis in urban Dar es Salaam, Tanzania. Parasites & Vectors. 2015;8(1):1-9.

4) Wagman JM, Grieco JP, Bautista K, Polanco J, Briceño I, King R, et al. The field evaluation of a push-pull system to control malaria vectors in Northern Belize, Central America. Malaria Journal. 2015;14(1):1-11.

5) Lynch P, Grimm U, Thomas M, Read A. Prospective malaria control using entomopathogenic fungi: comparative evaluation of impact on transmission and selection for resistance. Malaria Journal. 2012;11(1):383.

6) Read AF, Lynch PA, Thomas MB. How to Make Evolution-Proof Insecticides for Malaria Control. PLoS Biology. 2009;7(4):e58.

7) Cator L, Lynch P, Thomas M, Read A. Alterations in mosquito behaviour by malaria parasites: potential impact on force of infection. Malaria Journal. 2014;13(1):164.

[Editors' note: further revisions were requested prior to acceptance, as described below.]

*1) In the second paragraph of the Modelling section it should say "see Appendix 1".*

Corrected.

*2) Appendix 2 makes it clear that [dr], [dR] etc. refer to proportions of alleles in gametes. Shouldn't the Modelling section also make this clear?*

Main text amended, now reads “With [dr], [dR], [Dr], [DR], [d–] and [D–] representing, in the zygote genotypes for the population at a given time point, the proportion of alleles at the deflection locus which are non-deflection alleles paired with susceptible alleles, non-deflection alleles paired with resistant alleles, deflection alleles paired with susceptible alleles, deflection alleles paired with resistant alleles, non-deflection alleles paired with any resistance allele and deflection alleles paired with any resistance allele, respectively, assuming the same proportions in gametes of mating males and newly-emerged females.”

*3) Near the end of the Modelling section it contains a couple of "it can be seen" statements that seem either incomplete, imprecise or require further justification. In the first it claims that spread of deflection will by favoured by maximising the "fitness difference between susceptible and deflected phenotypes", but this won't be true if F_D_ < F_S_.*

The ESR concept inherently depends on deflected phenotypes having higher fitness than susceptible (not deflected not resistant) phenotypes, i.e. the fitness cost of entering properties and being killed in properties treated with insecticide must be higher than the fitness costs of being deflected away from properties treated with repellent. We had not stated this clearly in the text, and have now amended the text to correct this. (Modelling section, third paragraph from the end).

*Also not clear why there is no reference here to the F_R_ – F_D_ term.*

We have amended the text to explicitly state that maximising the fitness of deflected relative to resistant phenotypes will support establishment of an ESR.

*The second "it can be seen" is even less clear as there are terms on the rhs of the inequality that both increase and decrease with increasing frequency of resistance alleles. I'm prepared to believe it's generally true with plausible parameters, but I don't think it's necessarily easily seen.*

We have reviewed the ‘it can be seen’ statements and made amendments to the text to try to make things clearer and more precise. We have relocated an explanation linking the recommended application strategy to the fitness of deflected resistant phenotypes to a later point in the text where this has a clear and simple impact on the spread of resistance if deflection reaches fixation.

*4) In the fourth paragraph of the Modelling section "data become" rather than "data becomes" (also some slightly erratic punctuation throughout which no doubt will be fixed in the production process).*

Corrected.

*5) In the fifth paragraph of the Results "0.05%" – the Figure 2 caption says 0.5% – presumably the latter is correct.*

Corrected, with thanks to the reviewer for spotting this.